# Analysis of UAS-LiDAR Ground Points Classification in Agricultural Fields Using Traditional Algorithms and PointCNN

**Nadeem Fareed [1], Joao Paulo Flores [1,*] and Anup Kumar Das [1]**

Department of Agricultural and Biosystems Engineering, North Dakota State University, Fargo, ND 58102, USA
* Correspondence: paulo.flores@ndsu.edu; Tel.: +1-701-231-5348

**Abstract:** Classifying bare earth (ground) points from Light Detection and Ranging (LiDAR) point clouds is well-established research in the forestry, topography, and urban domains using point clouds acquired by Airborne LiDAR System (ALS) at average point densities ($\approx$2 points per meter-square (pts/m$^2$)). The paradigm of point cloud collection has shifted with the advent of unmanned aerial systems (UAS) onboard affordable laser scanners with commercial utility (e.g., DJI Zenmuse L1 sensor) and unprecedented repeatability of UAS-LiDAR surveys. Therefore, there is an immediate need to investigate the existing methods, and to develop new ground classification methods, using UAS-LiDAR. In this paper, for the first time, traditional ground classification algorithms and modern machine learning methods were investigated to filter ground from point clouds of high-density UAS-LiDAR data ($\approx$900 pts/m$^2$) over five agricultural fields in North Dakota, USA. To this end, we tested frequently used ground classification algorithms: Cloth Simulation Function (CSF), Progressive Morphological Filter (PMF), Multiscale Curvature Classification (MCC), and ArcGIS ground classification algorithms along with the PointCNN deep learning model were trained. We investigated two aspects of ground classification algorithms and PointCNN: (a) Classification accuracy of optimized ground classification algorithms (i.e., fine adjustment is user-defined parameters) and PointCNN over training site, and (b) transferability potential over four yet diverse test agricultural fields. The well-established evaluation metrics of omission error, commission error, and total error, along with kappa coefficients showed that deep learning outperforms the traditional ground classification algorithms in both aspects: (a) overall classification accuracy, and (b) transferability over diverse agricultural fields.

**Keywords:** UAS; LiDAR; point clouds; precision agriculture; PointCNN; CSF; PMF; MCC

## 1. Introduction

Light Detection and Ranging (LiDAR) sensors have been commercialized among growing remote sensing technologies, spreading over all major scales and platforms such as ground-based terrestrial laser scanners (TLS), vehicle-based mobile laser scanners (MLS) inclusive hand-held devices, unmanned aerial systems (UAS) laser scanners, or an airborne (manned) laser scanner (ALS) and spaceborne land elevation satellites deciphering three-dimensional (3D) objects with centimeter to millimeter-level accuracies [1–4]. In comparison with all the major platforms, UAS is highly customizable ensuring several payloads during a single mission such as LiDAR and red-green-blue (RGB) optical sensors commonly referred to as digital aerial photogrammetry (DAP) [5–7]. UAS-DAP provides multispectral information for LiDAR point clouds as most LiDAR sensors operate in the infrared portion of the electromagnetic spectrum with a single channel (e.g., 1064 nm) [8]. Additionally, UAS-DAP is capable of generating point clouds through structure-from-motion (SfM) techniques [9]. Compared with UAS-LiDAR, UAS-DAP point clouds can only measure the top of the canopies; therefore, significant information loss occurs below the canopy. In addition, optical sensors are highly sensitive to light conditions, shadow, and occlusion effects [10]. The occlusions are the most fundamental problems with remotely sensed data.

The mechanisms are classified as absolute, geometric, and turbid occlusions [11]. The absolute occlusion is caused by solid objects, for example, rooftops, tree trunks, branches, etc. Geometric occlusions are caused by directional blocking due to nadir off-nadir positioning of imaging/scanning sensors. Finally, turbid occlusions are least in context and are caused by the medium through which light travels; therefore, weather-dependent mechanisms can be avoided. LiDAR sensors mainly suffer from absolute occlusion (e.g., tree trunk and branches), because laser penetration capabilities, therefore, overcome geometric and turbid occlusions to better present the 3D volumetric presentation, e.g., height, canopy dimensions, gaps, and biomass of trees and crops compared with UAS-DAP [11–13].

Additionally, compared with manned ALS, UAS-LiDAR allows far-denser point clouds over low-cost small-area projects with easy operation and faster post-processing, as deliverables are available within a few hours post-survey. Moreover, UAS-LiDAR offers unprecedented repeatability compared with expensive wide-area ALS surveys [5]. UAS-LiDAR integrated with Global Navigational Satellite System (GNSS) or Real Time Kinematics (RTK) and Inertial Measurement Unit (IMU) transmit the geolocated laser pulses and record the backscattered signal in the full waveform or more widely adopted three-dimensional (3D) cloud of discrete laser measurements known as point clouds [14]. UAS-LiDAR point clouds are geolocated with precise horizontal measurements ($x$, $y$) with elevation ($z$) above mean-sea-level (MSL) with an average centimeter-level accuracy [4,5,15]. The strength of the backscattered LiDAR signal referred to as intensity information is also provided in most of the LiDAR data. The laser backscattered signal from bare earth is referred to as ground points and those off the terrain (OT), e.g., trees, plants, and buildings, are called non-ground points [16]. Nevertheless, point clouds generated by processing GNSS or RTK, IMU, and LiDAR datasets have no prior discrimination of ground and non-ground points. To use point clouds for accuracy assessment or to produce Digital Elevation Models (DEM), Canopy Height Models (CHM), and Crop Surface Models (CSM), highly accurate ground point classification is the most important step in LiDAR data processing [15]. Manual quality control (QC) and ground classification have been found most accurate, yet they are slow and labor-intensive. With the advent of UAS-LiDAR, more and more point clouds are available with an ever-increasing demand for ground point classification.

### 1.1. Related Work

For ground classification, elevation ($z$) is widely used information that accounts for anything elevated above the ground, therefore establishing the basis for ground classification algorithms [17]. Based on points elevation ($z$), algorithms categorically fall into four classes [18], segmentation/clustering [19], morphology [20], Triangle Irregular Network (TIN) [21], and interpolation [22]. Over the last two decades, ALS has extensively been adopted in earth sciences and, in particular, in investigating topography and forestry [23,24]. In forests and topography, the preponderance of research investigations was dedicated to classifying ground points from non-ground points to develop DEM, CHM, and CSM over state or national scales [24]. Compared with traditional remote sensing methods and optical imagery, ALS offers unprecedented topographic detail and precision over complex terrains. Nevertheless, the ALS point density (points per square meter area (pts/m$^2$)) varies greatly with the sensor's operational parameters such as flying altitudes, pulse repetition rate (PRR), and pulse penetration rate (PPR) [25]. In a typical ALS survey, point density could range from 1 to 15 pts/m$^2$ and the latest ALS system could achieve a maximum of 204 pts/m$^2$ [26]. ALS has been a widely adopted technique for decades; almost all the ground classification algorithms were developed using low-density evenly distributed point clouds with an average of 2 pts/m$^2$ [27]. Compared with ALS, UAS-LiDAR point densities have increased several fold in recent times, e.g., 1630 pts/m$^2$ with an average of 335 pts/m$^2$ [12]. Furthermore, based on cost, UAS-LiDAR sensors are classified as high-accuracy high-end, and affordable low-cost sensors with lower accuracy as compared to high-end LiDAR sensors, offering point clouds of varying degrees of precision with

complex geometries of objects under investigation [28]. Consequently, UAS-LiDAR data are comprised of richer geometric information compared with low-density ALS data [5,29].

In recent times, several studies were based on traditional ground classification algorithms using UAS-based point clouds. Zhou et al. (2022) used UAS-DAP and UAS-LiDAR point clouds to classify ground and non-ground points using the CSF algorithm in urban environments [10]. Moudrý et al. (2019) classified ground and non-ground points of a post-mining site using Riegl LMS-Q780 full-waveform airborne LiDAR and UAS-DAP point clouds with ArcGIS 10.4.1 software application (ESRI, Redlands, CA, USA) [30]. In another study, Brede et al. (2019) estimated tree volumes using UAS-LiDAR in forest environments where the ground points were manually labeled using CloudCompare 2.10 (http://cloudcompare.org/ (accessed on 4 January 2023)) software application [31]. In mangrove forest monitoring, Navarro et al. (2020) used the PMF algorithm to classify ground points of UAS-DAP point clouds [32]. Zeybek and Şanlıoğlu (2019) used open-source and commercial software packages to assess the ground filtering methods using UAS-DAP point clouds in landslide-monitoring forested research areas [33]. The previous studies highlight several important findings. First, ground classification offers a wide variety of algorithms where most authors used a single algorithm without any logical conclusion of their choice [10,30]. Some authors preferred a manual approach to that of an algorithm-based approach to ensure the high accuracy of classified ground points [31]. Nevertheless, Zeybek and Şanlıoğlu's (2019) study was based on an inter-comparison of four different ground classification algorithms, which is potentially relevant to UAS-DAP ground points classification rather than UAS-LiDAR point clouds [32]. In addition, the transferability evaluation of ground classification algorithms to other test sites has not been conducted in their research. Additionally, intra-comparison of traditional ground classification algorithms with modern deep learning methods is lacking in the past studies [30–33].

In an agricultural environment, rigorous evaluation of ground classification using UAS-LiDAR point clouds is pending. Nevertheless, the accurate classification of UAS-LiDAR ground points in an agricultural environment is an important processing step to monitor the plant height, stem size, crown area, and crown density during the growth stages. UAS flights are frequent throughout the growing season resulting in massive point clouds. Moreover, during the growing season UAS-LiDAR surveys, the overall crop structure, and point densities do not remain the same. Henceforth, the automated and precise classification of ground points of UAS-LiDAR data is in great demand for various stakeholders in agricultural domains. In the past, the difficulty of ground point classification was considered a major issue compromising the accuracy assessment of grassland structural traits [12,34].

### 1.2. Contributions

To the best of our knowledge, scientific investigations on ground point classification using UAS-LiDAR point clouds have not yet been conducted. The overarching goal of the present study is to reach a robust and automated solution for ground classification using UAS-LiDAR point clouds acquired in agricultural fields. Our objective is to assess the performance of the frequently used ground filter algorithms [17,20,22,35] such as (a) Cloth Simulation Function (CSF), (b) Progressive Morphological Filter (PMF), and (c) Multiclass Curvature Classification (MCC) all available in open-source R programming [36] using a similar approach of Zeybek and Şanlıoğlu (2019) [32]. Additionally, the ground classification algorithms implemented in Environmental System Research Institute (ESRI) ArcGIS need to be assessed because ArcGIS is a frequently used software package available and utilized in academia and industry [17,27]. Additionally, Deep learning (DL) has proven to be robust in point cloud segmentation, and classification using convolutional neural networks (CNNs) [37,38]. In the context of ground classification, we aim at using the PointCNN model [39] as a modern tool compared with traditional algorithms to classify ground points using UAS-LiDAR point clouds [16,40]. Finally, we aim at assessing the transferability potential of the traditional ground classification algorithm along with

DL methods, which is an extension to the approach proposed by Zeybek and Şanlıoğlu (2019) [32].

The rest of the paper is structured as follows. Study design along with representative agricultural plots description, data collection, and post-processing are explained in Section 2. The methodology based on the ground filtering algorithms, PointCNN framework, and along with mathematical formulation of error metrics is described in Section 3. Section 4 is dedicated to the result assessment in the context of qualitative and quantitative analyses, and Section 5 is based on the discussion of the results. Finally, the study is concluded in Section 6 by highlighting the significant findings.

## 2. Study Design and Materials

The present study comprised UAS-LiDAR data collection and post-processing over five experimental sites (Figure 1a) using automated software applications. Following the data processing, the optimization of ground classification algorithms, training, and valida-tion of the PointCNN model over training Site A (Figure 1b). Finally, the transferability assessment of the optimized ground classification algorithm and trained PointCNN model over four representative test sites is performed (Figure 1c).

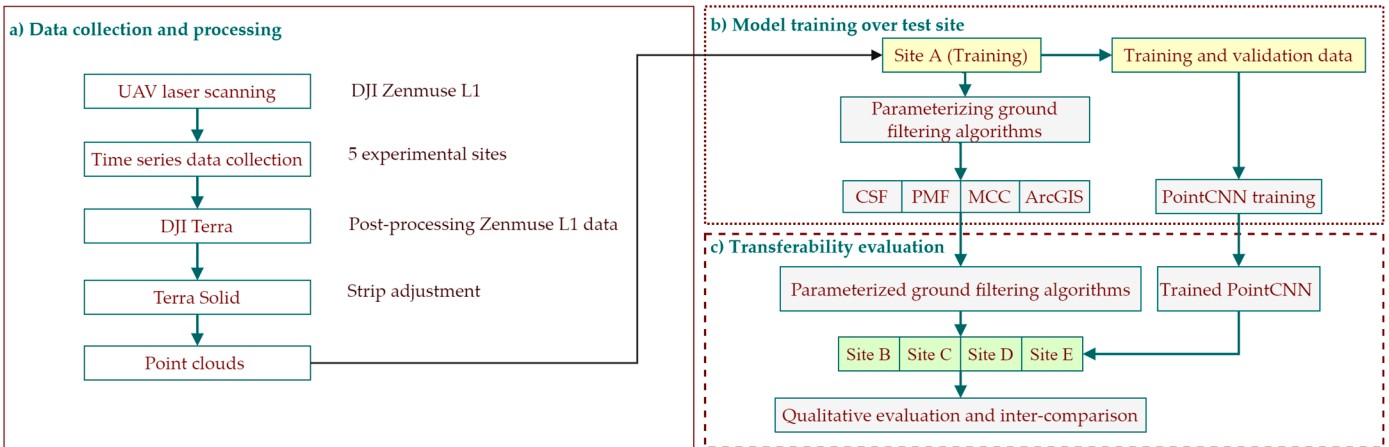

**Figure 1.** The schematic study designs: (**a**) Data collection and processing. (**b**) Parameter's opti-mization of ground classification algorithms and training PointCNN over Site A. (**c**) Transferability evaluation of ground classification algorithms and PointCNN over representative test sites.

### 2.1. Study Sites

UAS-LiDAR datasets were collected over training site A and four test sites (B, C, D, and E), throughout the growing season (July to August 2022). The agricultural fields comprised of different sizes, crops, planting patterns, planting densities, and growth stages (Table 1) were selected. The training (Site A), and test sites B, C, D, and E, were spread across different locations in North Dakota, USA, as shown in Figure 2.

**Table 1.** The descriptive statistics of study sites in North Dakota, USA.

| ID | Site/Field | Crop | Area (m$^2$) | Description | Crop Stage |
|----|-----------|------|---------|-------------|------------|
| A | Grand Farm | Corn, soybean, sunflower, sugar beet | 6773 | Training site | Late season |
| B | Leonard | Soybean | 6624 | Test site | Early season |
| C | Grand Forks | Pulse, Peas, wheat | 3312 | Test site | Mid-season |
| D | Minot | Pulse | 11,344 | Test site | Mid-season |
| E | Carrington | Wheat | 13,312 | Test site | Mid-season |

### 2.1.1. Grand Farm—Site A

The training Site A covers an area of 6773 m$^2$ and is located south of Fargo, North Dakota, USA (Figure 2), and is comprised of dry corn, green corn, sunflower, sugar beet, and soybean at the crop stage of the late season with stagnant growth. In addition, a significant portion of bare earth partly covered with weeds and grass is available in Site A. Therefore, Site A offers the desired complex crop environment to collect the experimental UAS-LiDAR data for the assessment of traditional ground classification algorithms in comparison with deep learning methods (e.g., PointCNN). For example, dry corn has more under-canopy returns compared to green corn offering geometrically diverse UAS-LiDAR point clouds, as demonstrated in Figure 2a (see transects A1 and A2).

### 2.1.2. Leonard—Site B

Test Site B is of an area of 6624 m$^2$ is located south of the Grand Farm and consists of pea-crop at early growth stages with a plant size of fewer centimeters as shown with transect B1 in Figure 2b. Site B was chosen to assess the optimized ground classification algorithms and PointCNN performance for the growing crops at early season stages compared with a nearby significantly matured crop, as shown in Figure 2b (see transects B1 and B2). Ground classification in an environment with small plants and shrubs was found to be challenging for LiDAR data in past studies [27].

### 2.1.3. Grand Forks—Site C

Grand Forks test site is of an area of 6624 m$^2$ located north of Fargo, North Dakota, USA (Figure 2), and is mostly covered with the pulse, pea, weed, and grass at active growth stages offering more complex UAS-LiDAR point clouds for mid-season crops, as shown in Figure 2c (see transects C1 and C2).

### 2.1.4. Minot—Site D

Site D is comprised of a vast area of 11,344 m$^2$ located northwest of Fargo, North Dakota, and is covered with dry soybean with undulating terrain, along with vegetated ditches in agricultural plots as shown in Figure 2d (see transects D1 and D2). Site D with vegetated ditches offers a unique testing environment to assess the ground point classification performance for an undulating topography.

### 2.1.5. Carrington—Site E

Site E consists of a large area of 13,312 m$^2$ located northwest of Fargo, North Dakota, containing a large portion of logged wheat along with pulse, pea, grass, and bare-earth; therefore, it qualifies, producing a unique UAS-LiDAR test dataset, as shown in Figure 2e (see transects E1 and E2).

### *2.2. UAS-LiDAR Data Collection and Post-Processing*

UAS-LiDAR Surveys of study areas (Figure 2) were conducted using a DJI Zenmuse L1 sensor onboard DJI Matrice 300 RTK with 50% forward and lateral overlap between two successive flight paths [41,42]. ZENMUSE L1 with a field of view of 70.4° horizontal and 4.5° vertical capable of generating 240,000 points/second at a maximum flying altitude of 450 m above ground level (AGL) [43]. DJI ZENMUSE L1 detailed sensor-related specifications can be found at [41]. The colored point clouds with RGB values from optical imagery in the American Society for Photogrammetry and Remote Sensing (ASPRS) LAS version 1.4 [44] were obtained by post-processing RTK, IMU, optical imagery, and laser scanning datasets using manufacturer-provided DJI Terra software application [45]. In a UAS-LiDAR survey, an area of interest is generally covered with several parallel overlapping flight paths, e.g., green, red, and yellow strips in Figure 3c. A fixed overlap is maintained to avoid data voids due to the occlusions effect [11]. Nevertheless, a typical UAS-LiDAR system is of multi-sensors integration comprised of a laser scanner, an IMU, and a GNSS or RTK) with some degree of error probability referred to as "systematic

error". The systematic error of the position and orientation system (POS) somewhat offsets the same object in the adjacent overlapping flight paths (see Figure 3a, transect A′A″ in Figure 3b) [46]. To ensure the LiDAR data consistency and quality, the offsets among the UAS-LiDAR data within overlapping flight paths (Figure 3b) need to be corrected [46]. In this study, we used TerraSolid (TerraSolid, Ltd., Jyväskylä, Finland) software application (https://terrasolid.com/products/terrasolid-uav/ (accessed on 1 January 2023)) for the UAS-LiDAR automated strip adjustment (see Figure 3c–d) [47].

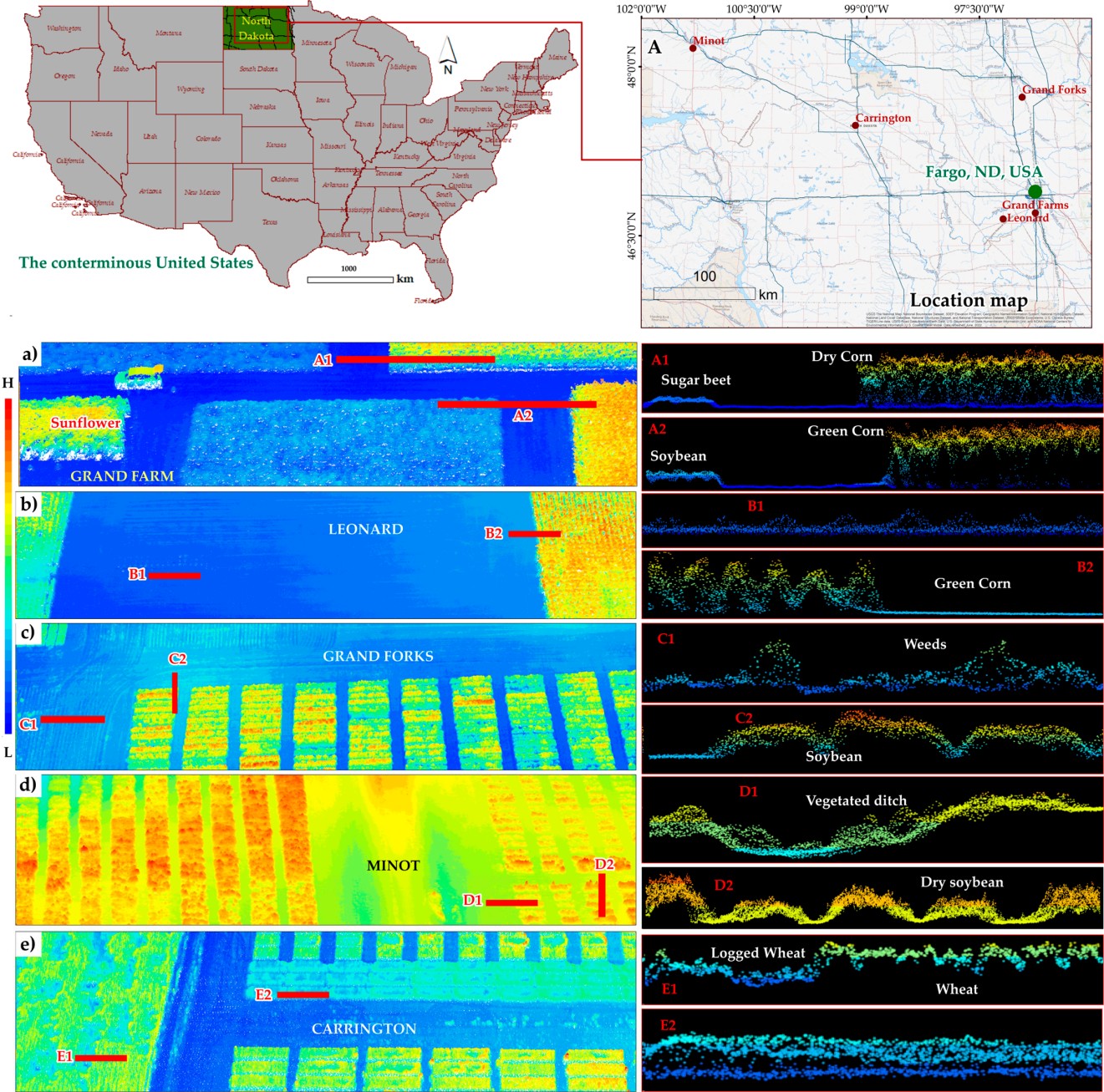

**Figure 2.** The study site's locations are in North Dakota (ND), USA (see site location map, top left): (**A**) zoom in view of representative sites in Fargo, ND, USA. (**a**) Grand Farms training site A. (**b**) Leonard, (**c**) Grand Forks, (**d**) Minot, and (**e**) Carrington are the test sites B, C, D, and E (Table 2). (**A1**–**E2**) are some characteristics3D-transects of UAS-LiDAR point clouds. Three-dimensional transects of (**a**–**e**) are of the same extent (horizontal) for visualization purposes with no vertical scale.

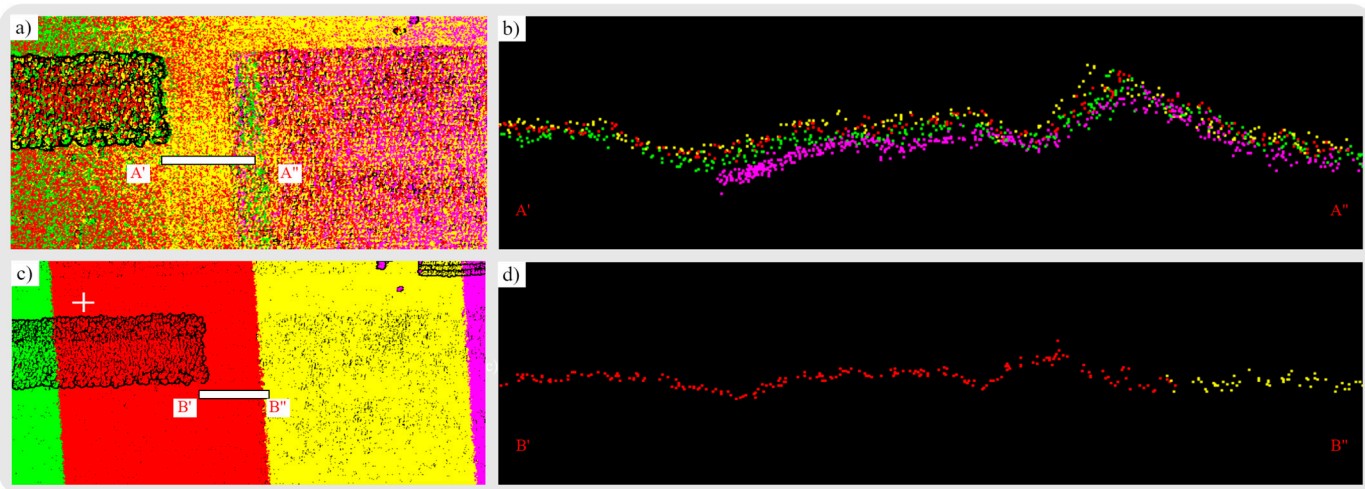

**Figure 3.** Strip adjustment of DJI L1 sensor: (**a**) Offset and overlapping UAS-LiDAR point clouds strips, (**b**) transect A′A″ before strip adjustment. (**c**) Strip adjusted point clouds, and (**d**) transect B′B″ after strip adjustment. Three-dimensional transects (**b**–**d**) are of the same extent (horizontal) for visualization purposes with no vertical scale.

The UAS-LiDAR surveys were carried out at different dates for each site with varying altitudes with a minimum of 30 m to a maximum of 50 m above-ground level (AGL) with pulse repetition rate (PRR) of 250 kHz or 160 kHz resulting in millions of laser returns comprised of a wide range of point densities of 590.82, 390.43, 593.5, 903.89, and 484 for training and test sites as shown in Table 2. The detailed descriptive statistics of acquired UAS-LiDAR point clouds over five representative sites are given in Table 2.

**Table 2.** Summary of collected Zenmuse L1 UAS-LiDAR datasets for five sites.

| ID | Survey Date | Altitude (m) | Frequency (kHz) | Area (m$^2$) | Pts/m$^2$ | Total pts (Million) |
|----|-------------|--------------|-----------------|--------------|-----------|---------------------|
| A | 08/22/2022 | 50 | 250 | 6773 | 590.82 | 4 |
| B | 07/21/2022 | 45.7 | 160 | 6624 | 390.43 | 2.5 |
| C | 07/15/2022 | 45.7 | 160 | 3312 | 593.5 | 1.97 |
| D | 08/22/2022 | 30.5 | 160 | 11,344 | 903.89 | 10.2 |
| E | 07/20/2022 | 45.7 | 250 | 13,312 | 484.5 | 6.4 |

### 2.3. Training Validation and Benchmark Point Clouds

The DL framework requires sufficient quality labeled data for training, validation, and testing [48]. For Point CNN, the training data represented with green tiles in Figure 4, and validation data with red tiles in Figure 4, each with dimensions 4 m × 4 m, were randomly selected from Site A. The points were manually labeled as ground and non-ground points using ESRI ArcMap LAS dataset profile view (https://desktop.arcgis.com/en/arcmap/latest/manage-data/las-dataset/las-dataset-profile-view.htm (accessed on 6 December 2022)) depicted with brown and green colors in Figure 2, respectively. To generalize the PointCNN, sufficient training and validation points were labeled as ground, and non-ground, for tall-sized crops (Figure 4a), medium-sized crops (Figure 4b), small-sized weeds and grass (Figure 3c), and only ground points (Figure 4d). The training, validation, and benchmark point clouds represent all crops including sugar beet, sunflower, green corn, dry corn, weeds, and ground points.

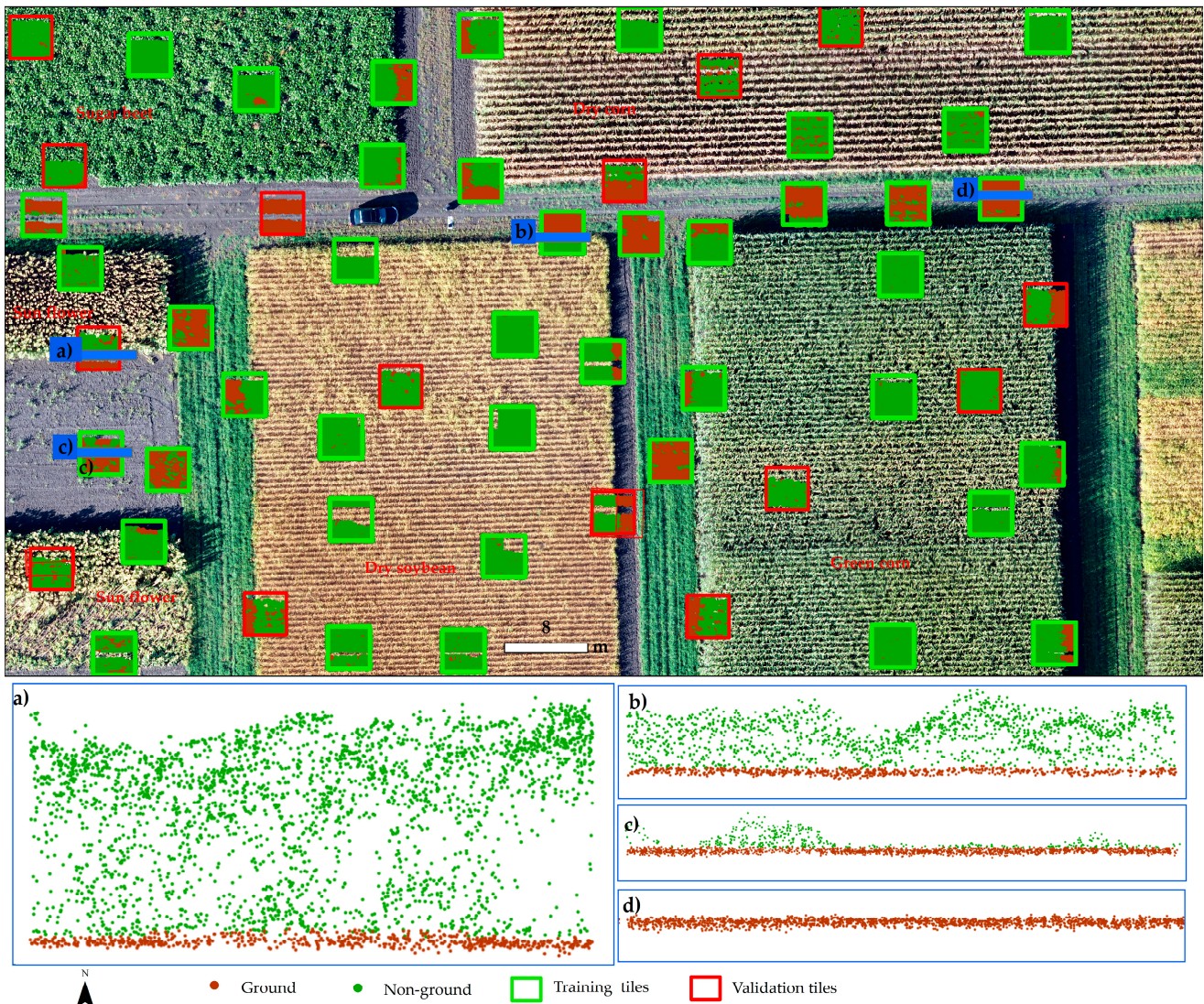

**Figure 4.** Site A. Example of training, validation, and benchmark data. Three-dimensional transects of point clouds training data of (**a**) sunflower, (**b**) soybean, (**c**) ground with some weeds, (**d**) ground. Three-dimensional transects (**a–d**) are of the same extent (horizontal) for visualization purposes with no vertical scale.

Figure 4 also depicts the approach used to label benchmark ground and non-ground points for training and test study sites (Figure 2). We aim to use benchmark points for the qualitative and quantitative analysis of the ground classification results produced by ground classification algorithms and PointCNN. The detailed descriptive statistics of training, validation, and benchmark points for training and test sites are provided in Table 3.

**Table 3.** Summary statistics of training, validation, and benchmark point clouds for study sites.

| Site ID | Category | Ground Pts | Non-Ground Pts |
|---|---|---|---|
| A | Training | 143,203 | 23,7659 |
| | Validation | 30,868 | 11,9743 |
| | benchmark | 14,769 | 24,684 |
| B | benchmark | 19,138 | 3720 |
| C | benchmark | 11,289 | 18,673 |
| D | benchmark | 25,812 | 57,128 |
| E | benchmark | 47,365 | 65,551 |

## 3. Methods

### 3.1. Ground Classification Algorithms

#### 3.1.1. CSF

The CSF algorithm first inverts the original point clouds upside-down, and then a cloth is simulated over the inverted point clouds (see Figure 5). Following the cloth simulation, cloth particles are distributed over the simulated cloth surface as depicted with red dots in Figure 5. The cloth particles carry a constant weight, which bends the simulated cloth surface under the influence of gravity following Newton's Second Law of motion. Finally, those points stick with the simulated cloth and then are classified as ground points [35]. The CSF algorithm is widely adopted for ground point classification and implemented in several software applications [36,49,50]. The best performance is subject to fine-tuning the default parameters [35,51]. In addition, the performance is subject to the study area under investigation, e.g., flat, or undulating terrain. As shown in Figure 5b, CSF has several default parameters such as grid resolution (GR) or cloth resolution, i.e., number of cloth particles, and rigidness (RI) of simulated cloth, i.e., the tension in the simulated surface. The value of 1 puts less tension in cloth, and cloth particles tend to bend more on the simulated surface, e.g., Figure 5b, location C, while higher values (e.g., 3) induce more tension in simulated cloth, which results in less bending, as shown in Figure 3b, and more suitable in flat agricultural fields [35]. Moreover, the cloth simulation is weak at steep slopes and therefore slope parameters should be considered for hilly terrain. Finally, the number of iterations were performed to fine-tune the cloth over inverted point clouds. In total, there are six default adjustable default parameters of the CSF algorithm [35].

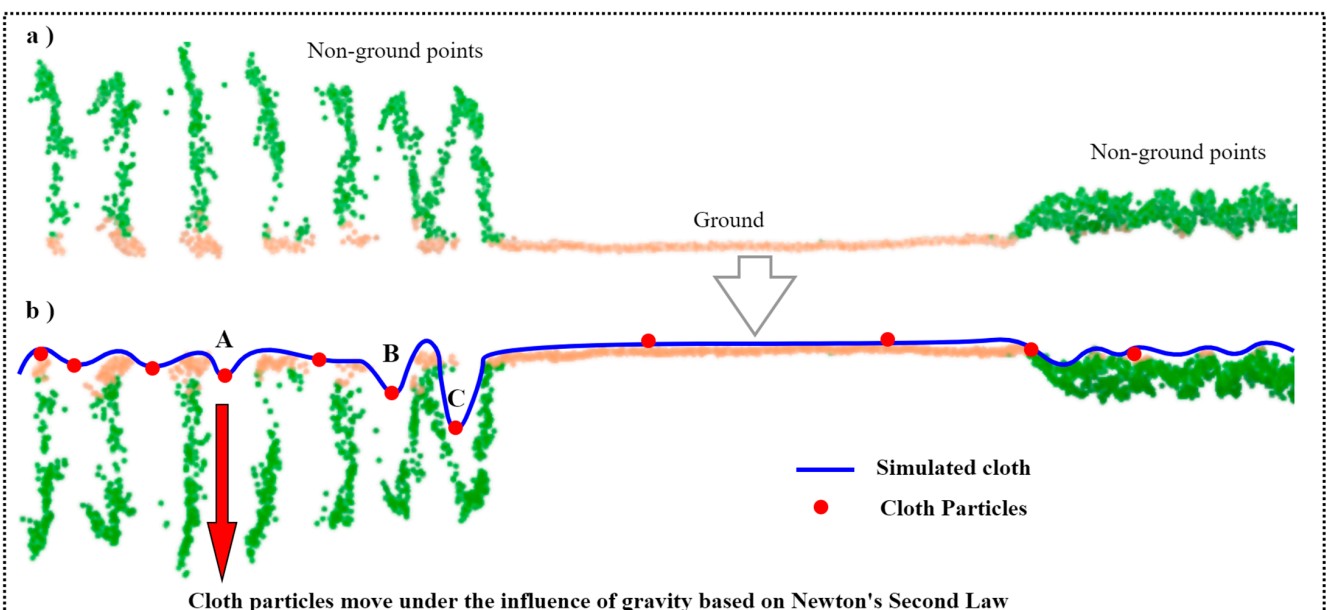

**Figure 5.** The CSF algorithm: (**a**) original point clouds. (**b**) Inverted point clouds with simulated cloth with cloth particles (red dots) with RI values 3 at A, 2 at B, and 1 at C, respectively.

#### 3.1.2. PMF

Morphological filters are based on mathematical morphology with two fundamental operations of dilation and erosion to extract features from images. The dilation process enlarges while the erosion process reduces the size of features in images [52]. In the context of LiDAR data processing, the point clouds are converted into a digital surface model (DSM), and dilation (Equation (1)) and erosion (Equation (2)) for elevation ($z$) at point ($x, y$) can be written as:

$$Dilation\ (D_p) = (x_p,\ y_p)\epsilon\ w.\max\ (z_p) \tag{1}$$

$$Erosion\ (E_p) = (x_p,\ y_p)\epsilon\ w.\min\ (z_p) \tag{2}$$



At any given point, $(x_p, y_p, z_p)$ are neighboring points within the filtering window ($w$). Equations (1) and (2) show that the dilation process retains the maximum elevation, and the erosion process retains the minimum elevation at a given point within the filtering window ($w$). The dilation and erosion result in two surfaces and points with minimum height differences (Δh) between two surfaces are considered ground points. For morphological filters, therefore, fixed window size ($w$) and height difference threshold (Δh) are the default parameters. The PMF is an advanced version of morphological filters; i.e., the window size is iteratively and automated increased until it reaches a user-defined window size ($w$) [40].

### 3.1.3. MCC

The MCC algorithm first begins with interpolating the elevations ($z$) of points using the thin-plate-spline (TPS) method [53] where the cell resolution is estimated by the scale parameter λ. TPS fitting using a variable window size with the minimum 12 nearest neighbor points is achieved and an invariant tension parameter (f = 1.5) is set across all the scale parameters λ. The interpolated raster surface, i.e., DSM, is then further processed using $3 \times 3$ mean kernels, resulting in a mean surface raster. Finally, curvature values (c) are generated using a mean surface raster. All the points meeting the conditional statement with point's elevation ($z$) > c are classified as non-ground points [22]. The curvature tolerance is a default parameter referred to as the tolerance parameter (t). Likewise, PMF and MCC require scale (λ) and curvature tolerance (t) with default values specified by the founding authors, i.e., λ = 1.5 and t = 0.3 [22].

### 3.1.4. ArcGIS

Ground classification algorithms implemented in ArcGIS pro are available with three choices. The first method is referred to as the "standard method", implemented for gradually undulating topography with tolerance for little slope variations. The second method, named "conservative", uses tighter restrictions for slope changes to consider the low-laying foliage of grass and shrubbery, thus qualifying for assessment in an agricultural landscape. The third method is known as "aggressive" and is suitable for high degrees of slope variations and therefore of little use in an agricultural context. Therefore, among the available ground classification methods in the ArcGIS pro application, "standard" and "conservative" methods were assessed for ground point classification. Unlike, CSF, PMF, and MCC, ArcGIS ground classification algorithms come with optional default parameters such as reusing existing ground points during ground classification, offering more automation but little control over the classification process. Consequently, point cloud processing using ArcGIS is a comprehensive proprietary black-box solution with an unknown underlying mathematical formulation [17].

### 3.1.5. Ground Classification Algorithms—Optimization

The point clouds were processed first using algorithm's default values as proposed by the founding authors. We aim to adjust the default values through the trial-and-error method by changing one or two parameters while keeping the other parameters constant; therefore, the term "optimization" is used. Finally, an extreme adjustment approach is also tested with all parameters adjusted at the same time based on the analyst's practical experience gained in previous steps. For CSF, GR is incrementally adjusted from 0.5 to an extreme value of 0.1. The recommended rigidness constant value of 3 is adopted for flat agricultural fields with constant iterations of 1000 [35]. A similar approach is followed for PMF and MCC algorithms. Point clouds of study areas (Figure 2) were processed using ground classification algorithms implemented in lidR package [36]. The lidR is an open-source R package for processing ALS data with the advanced functionality of splitting massive point clouds into manageable user-defined chunks or tiles [14], removing outliers or noise points, and ground classification. We automated the point cloud processing using

lidR package. First large point clouds were tiles of size 10 m$^2$ and then noise points from each tile were removed using a statistical outlier's removal (SOR) filter [54].

### 3.2. PointCNN

The hierarchical representations available in regular gridded data such as imagery are well learned using convolutional neural networks (CNNs) by leveraging spatially local correlation in images. On the contrary, point clouds represent irregularity with unordered point clouds distributed in 3D space where traditional CNN is ill suited [39]. PointCNN adopts a hierarchical representation approach for unordered point clouds by using the X-Conv operator. The X-Conv recursively operates in local regions (analogous to local patches in an image) referred to as "project" or "aggregate" where information from the neighboring points is dissolved into fewer representative points with richer information with an encoder–decoder paradigm [48]. Figure 6b,c depict the principle functioning of X-Conv operators that gradually transform the input dense point clouds. Green dots in Figure 6b and blue dots in Figure 6c with an increased number of channels show the representation sequences. The overall DL architecture of PointCNN with two X-Conv operators is shown in Figure 6d, where N represents output representative points, C stands for dimensionality, K is neighboring point numbers found through search radius or K nearest neighbors, and D is the dilution rate. The comprehensive study of PointCNN (https://github.com/yangyanli/PointCNN (accessed on 6 November 2022)) is found at [39].

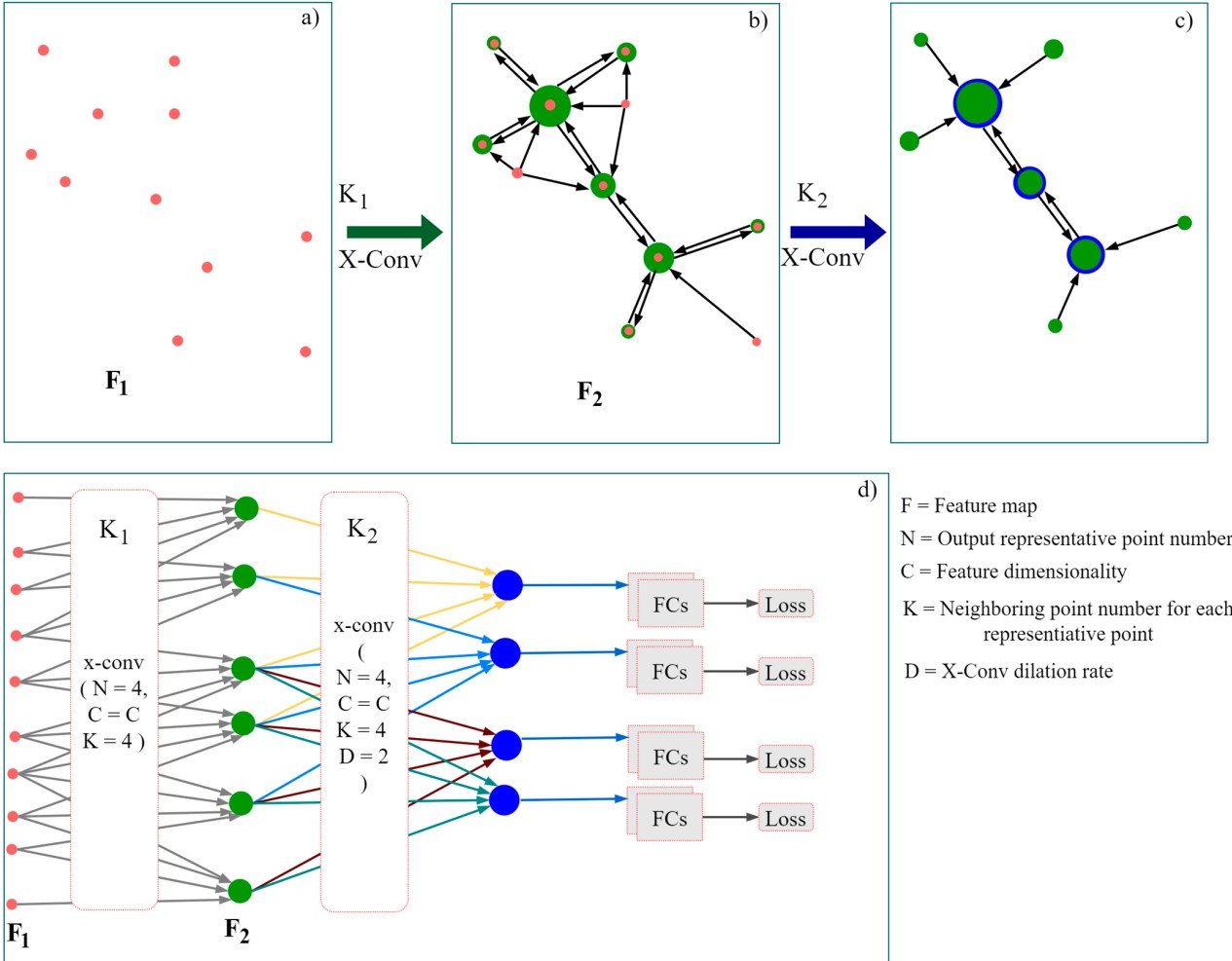

**Figure 6.** Principle functioning of PointCNN model: (**a**) Input dense point clouds, (**b**,**c**) diluted point clouds (green and blue dots) with fewer representations with multiple channels. (**d**) The overall Point CNN framework with two X-Conv operators.

In recent studies, PointCNN has been used for maze stem and leaf segmentation [55], agricultural scene classification [38], and extracting wood parameters from artificially planted forests [48]. However, extracting ground points using PointCNN has not been investigated. The ArcGIS DL framework provides the platform for preparing training and validation datasets along with model training. PointCNN is available under ESRI deep-learning framework (https://github.com/Esri/deep-learning-frameworks (accessed on 6 November 2022)) that works with ArcGIS Pro with a stand-alone installer "ProDeepLeearning.msi" [56]. Pro Deep Learning installer includes a collection of all major DL frameworks such as PyTorch (https://pytorch.org/ (accessed on 6 November 2022)), Fast.AI (https://www.fast.ai/ (accessed on 6 November 2022)), TensorFlow (https://www.tensorflow.org/ (accessed on 6 November 2022)), and Keras (https://keras.io/ (accessed on 6 November 2022)).

### 3.2.1. Training Validation and Validation Accuracy

To train PointCNN, the training and validation points (LAS 1.4 format) were converted into HDF5 [57] format using the ArcGIS Pro DL framework. A total of 729 training and 265 validation blocks each of size 4 m × 4 m were created. The details of the total training and validation blocks along with the distribution of total points in each block are shown in Figure 7. Using the training and validation data shown in Figure 7, the Point CNN was trained using Intel(R) Core (TM) i7-8700 CPU @ 3.20 GHz 3.19 GHz, with installed NVIDIA GeForce GTX 1080 graphics processing unit (GPU) and 64 gigabytes (GB) random access memory (RAM). Through the trial-and-error approach, the batch size of 4 has proven to be effective with installed memory of 64 GB. PointCNN was trained using the One Cycle Learning Rate (e.g., the learning rate for each epoch is recalculated using Fast.AI implementation) approach, which improves the CNN training and automatically finds the optimal learning rate at each epoch [58].

For 100 epochs, a total of 18,300 iterations were performed with 729 training and 265 validation blocks with an overall learning rate of 0.003 for two classes, i.e., ground and non-ground points. The characteristics curves of training loss, validation loss, and overall validation accuracy are shown in Figure 8.

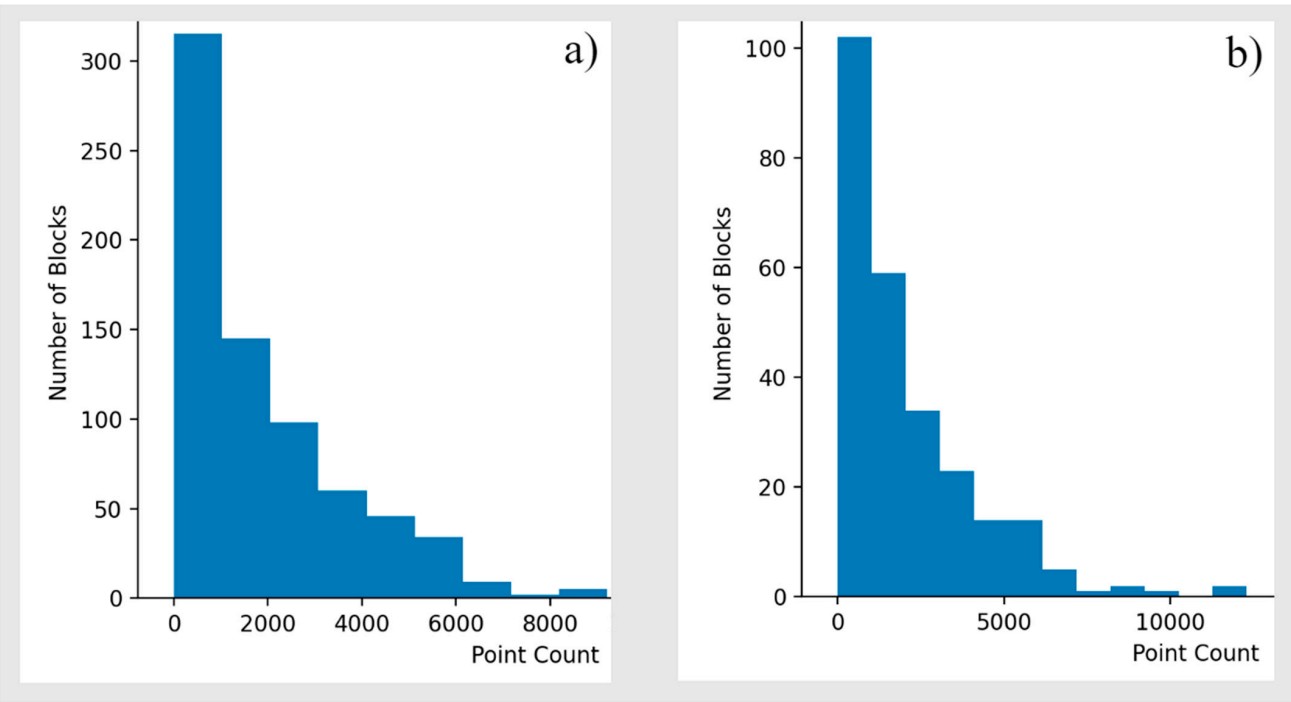

**Figure 7.** The distribution of HDF training and validation blocks of Site A: (**a**) Training data blocks. (**b**) Validation data blocks.

The best-trained model reaches an overall validation accuracy of 90% with a final training loss of 0.13 and a validation loss of 0.27, respectively (Figure 8). To assess the model performance over extended training periods; e.g., 200 and 150 epochs were also tested. We found no improvement in the model performance with extended training. The 10% validation accuracy loss is further discussed in Section 5.

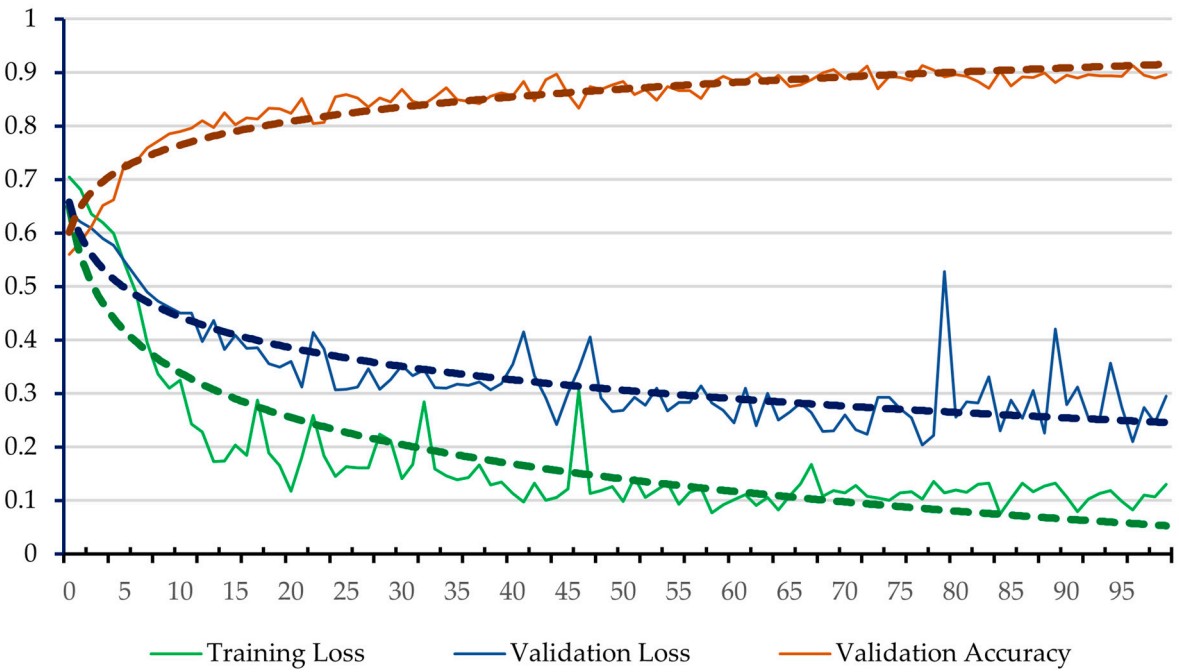

**Figure 8.** PointCNN train loss (green line), validation loss (blue line), and validation accuracy (orange line) for training parameters.

### 3.3. Quantitative Evaluation Metrics

The quantitative evaluation of ground classification algorithms and PointCNN were accomplished using a well-established approach [59]. Compared with benchmark points (Table 3), the number of ground points classified correctly are denoted with ($G_c$) and non-ground points ($N_c$) respectively. Type I (T.I.) error originates when a ground point ($G_c$) is classified as a non-ground point (($N_c$), e.g., error of omission denoted with $O_e$ in Equation (3)); a Type II (T.II.) error is caused when a non-ground point ($N_c$) is classified as ground (($G_c$), e.g., error of commission denoted with $C_e$ in Equation (4)); and Total error (T.E.) stands for total false points (e.g., omission $O_e$ and commission $C_e$ over total points ($T_p$) in a given sample (Equation (5)) [59]. The fourth statistical measure of the inter-ratio agreement metric, i.e., Cohen's kappa coefficient ((k), Equation (6)) found to be more robust for inter-comparison and therefore used in this study [60]. The respective equations of each metric are given below.

$$T.I.= O_e / (O_e + G_c) \tag{3}$$

$$T.II.= C_e / (C_e + N_c) \tag{4}$$

$$T.E. = (O_e + C_e)/T_p \tag{5}$$

$$\text{For } k \quad \begin{aligned} f &= (G_c + O_e)/ T_p & g &= (C_e + N_c)/ T_p \\ h &= (G_c + C_e)/ T_p & i &= (O_e + N_c)/ T_p \\ Pr(G_c) &= (G_c + N_c)/T_p & Pr(T_p) &= (f \times h) + (g \times i) \end{aligned}$$

$$k = (Pr(G_c) - Pr(T_p)/(1 - Pr(T_p)) \tag{6}$$

Point-to-point accuracy evaluation is more accurate but challenging and generally carried out manually for small datasets. Particularly for large datasets, point-to-point evaluation is labor-intensive, consequently, supported by interpolating the point clouds into raster data [21,49].

We introduced a novel method to automate point-wise omission and commission errors using algorithm-based point clouds (Figure 9B) compared with benchmark point clouds (Figure 9A). First, the non-ground points (benchmark, Figure 9a) and non-ground points (algorithm-based, Figure 9b) were merged into a single dataset (Figure 9c). Algorithm-based non-ground points that were filtered correctly share the same coordinates ($x$, $y$, $z$) as the benchmark non-ground point and therefore can be removed as duplicates from the merged dataset (Figure 9c); i.e., the duplicates as true match with the benchmark (i.e., valid points). For an ideal scenario, for example, zero-commission error, after removing duplicates, the total points must remain the same as the benchmark points. For a commission error in algorithm-based non-ground points (Figure 9B), after removing duplicates, the total number of non-ground points must exceed the original benchmark non-ground points (see yellow dots in Figure 9d). The additional non-ground points originate as commission error; i.e., some ground points were classified as non-ground points by an algorithm (compare Figure 9A,B,d). Following the same approach (Figure 9e–h), the surplus points in ground points after removing duplicates are commission errors of ground points; i.e., some non-ground points (e.g., red dots in Figure 9h) were filtered as ground points (see Figure 9A,B) by the algorithm. Figure 9 also indicates that the commission error of non-ground points (Figure 9d) is the omission error of ground points (Figure 9B). Using the proposed method, commission ($C_e$) and omission ($O_e$) errors of ground points were calculated to establish the evaluation metrics of T.I. (Equation (3)), T.II. (Equation (4)), T.E. (Equation (5)), and k (Equation (6)), respectively.

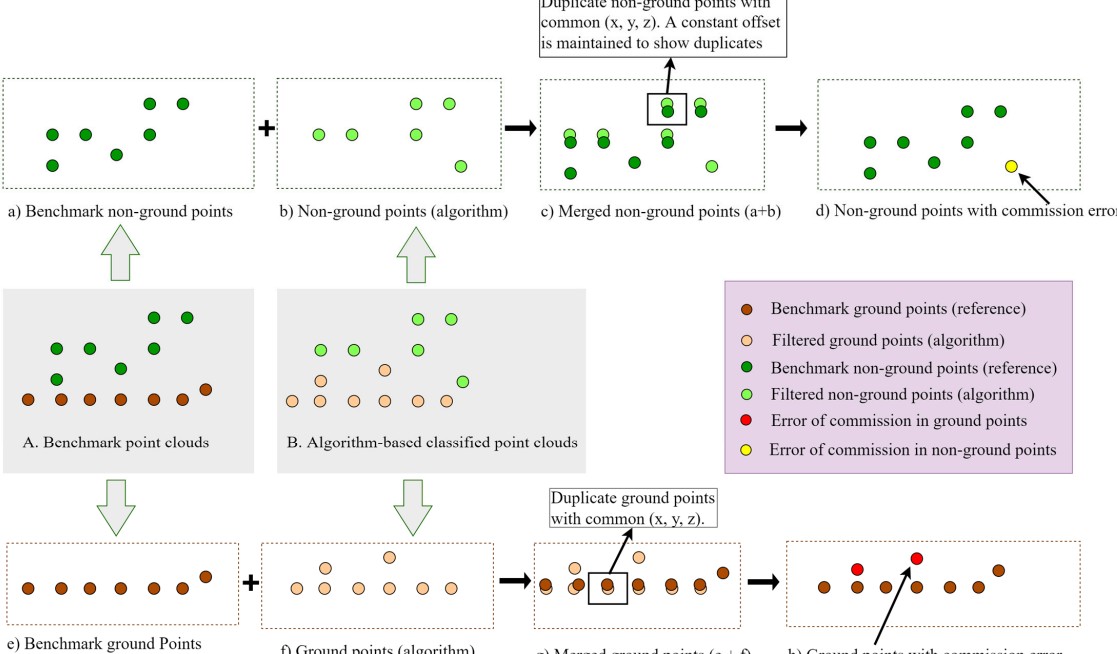

**Figure 9.** Automated workflow for calculating omission and commission errors in algorithm-based classified point clouds: (**A**) Benchmark points. (**B**) Algorithm-based classified points. (**a**) Benchmark non-ground points class. (**b**) Algorithm-based non-ground points. (**c**) merged non-ground points from (**a,b**) with duplicates. (**d**) Merged point clouds after removing duplicates. (**e**) Benchmark ground points. (**f**) Algorithm-based ground points. (**g**) Merged ground points with duplicates. (**h**) Ground points without duplicates.

## 4. Results

The results were first visually analyzed with qualitative analysis (Section 4.1) compared with the benchmark [14], and then quantitative analysis (Section 4.2) is established using the metrics of T.I., T. II., T.E., and k (Section 3.3), respectively.

### 4.1. Qualitative Analysis

Figure 10 shows the ground classification results of CSF, PMF, ArcGIS standard, ArcGIS conservative, and PointCNN compared with benchmark points. Among the ground classification algorithms, the MCC with default parameters values of λ = 1.5 and t = 0.30, the evaluation metrics of T.I., T.II., T.E., and k, were 04.24, 50.34, 27.34, and 45.16, respectively. Nevertheless, the MCC best-optimized parameters cannot be found through the trial-and-error for UAS-LiDAR data. MCC is found to be extremely slow with UAS-LiDAR data as it takes 8 h to process the 6773 m$^2$ area of Site A. The slow processing is possibly caused by thin plate spline (TPS) interpolation [16,40]. In addition, MCC suffers from tile-boundary artifacts [14] consistently for all adjustments in default parameter values. The tile-boundary artifacts affect the classification accuracy along the boundaries of point patches, as depicted with red and yellow boxes in Figure 10d. Likewise, ArcGIS ground classification algorithms were found to be poor in segregating ground and non-ground points over Site A. ArcGIS standard and conservative poorly perform at all distinct regions marked with yellow and blue boxes in Figure 10. As a result, the quantitative analysis of CSF, PMF, and PointCNN is further considered while MCC and ArcGIS were dropped from further analysis.

Among the ground classification algorithms available in the lidR package [36], CSF is found to perform better than PMF (Figure 10b–c), particularly, for low-lying sugar beet, as shown in Figure 10c. The visual assessment reveals that PointCNN is robust compared with CSF and PMF in terms of segregating ground from low-lying small grass or weeds, as shown with red bounding boxes in Figure 10a,g, respectively. Figure 10 indicates that most of the ground classification algorithms are more sensitive toward low-laying crops, as shown with yellow bounding boxes [61]. For tall crops such as corn shown with blue bounding boxes in Figure 10, the performance of CSF, PMF, and PointCNN is similar.

### 4.2. Quantitative Analysis

Tables 4 and 5 give the error metrics T.I., T.II., T.E., and k over-training Site A with the adjustment in default parameter values of CSF and PMF algorithms.

The CSF default parameters of dt, GR, and ts with values 0.50, 0.50, and 0.65 result in the error metrics of T.I., T.II., T.E., and k of 0.00, 51.18, 32.02, and 41.65, respectively (Table 4). The results indicate that with the authors provided default parameters, the CSF performs poorly with UAS-LiDAR data. Therefore, the optimized values were found by adjusting one or more default parameters. First, parameter-GR was sequentially adjusted in decreasing order from 0.50 to 0.10, as shown in Table 4. Finally, the optimized parameters, i.e., dt = 0.10, GR = 0.20, and ts = 0.30, were found by sequential adjustment in user-defined parameters (Table 4), which provided the least errors of T.I., T.II., TE., and k with values of 03.62, 05.76, 04.76, and 89.53, respectively. However, the extreme adjustment in default parameters by adjusting two or more default parameters at the same time, such as dt = 0.10, GR = 0.10, and ts = 0.20, resulted in T.I., T.II., T.E., and k values of 95.39, 0.004, 35.71, and 05.68 (Table 4), respectively. In the context of the parameter's sensitivity toward the error, dt is found more sensitive than GR as the GR variation from 0.50 to 0.30 improves the kappa coefficients from 41.65 to 49.19, while changes in dt from 0.30 to 0.15 resulted in k from 49.19 to 70.30, which is two times higher compared with a kappa coefficient of GR adjustment about the same magnitude.

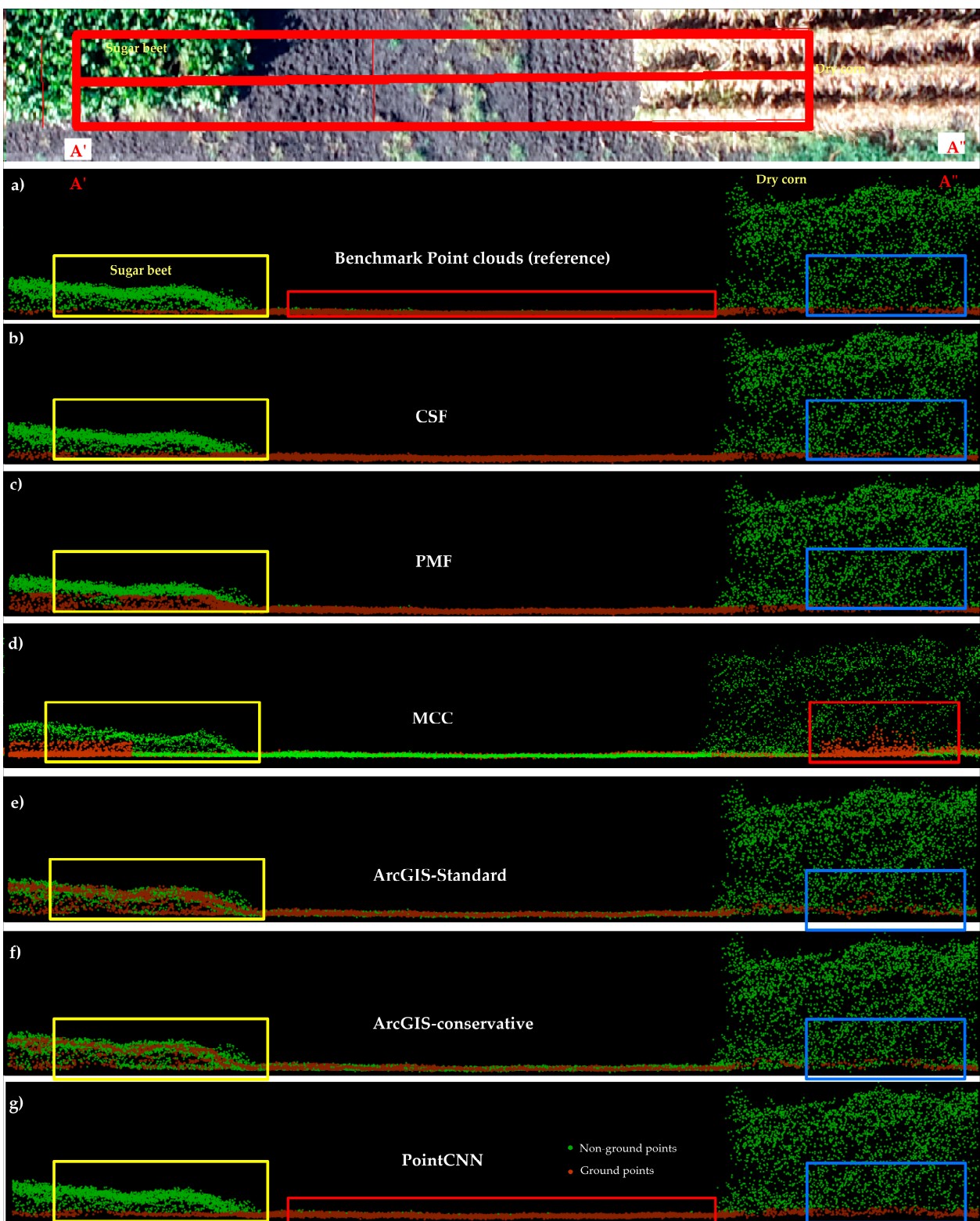

**Figure 10.** Visual evaluation (Site A) of ground classification algorithms and PointCNN using a 3D transect (A′A″) of point clouds: (**a**) Benchmark point clouds. Classification results for ground (brown) and non-ground (green) points using (**b**) CSF, (**c**) PMF, (**d**) MCC, (**e**) ArcGIS standard, (**f**), ArcGIS conservative, and (**g**) PointCNN. Three-dimensional transects (**a**–**g**) are of the same extent (horizontal) for visualization purposes with no vertical scale generated using ESRI ArcMap 10.8 LAS dataset profile view.

**Table 4.** CSF ground classification error metrics and kappa coefficients for different parameter values.

| Site ID | Method | Parameters | | | Results | | | |
|---------|--------|------|------|------|-----------|-----------|-----------|-----------|
| | | dt | GR | ts | T.I. (%) | T.II. (%) | T.E. (%) | k (%) |
| A | CSF | 0.50 * | 0.50 * | 0.65 * | 0.00 | 51.18 | 32.02 | 41.65 |
| | | 0.50 | 0.30 | 0.65 | 0.00 | 51.24 | 32.06 | 41.67 |
| | | 0.30 | 0.30 | 0.65 | 0.00 | 43.60 | 27.27 | 49.19 |
| | | 0.15 | 0.30 | 0.65 | 0.06 | 23.98 | 15.02 | 70.30 |
| | | 0.15 | 0.20 | 0.50 | 0.00 | 21.44 | 13.41 | 73.27 |
| | | 0.10 | 0.20 | 0.30 | 3.62 ** | 5.76 ** | 4.96 ** | 89.53 ** |
| | | 0.10 | 0.10 | 0.20 | 95.39 | 0.004 | 35.71 | 05.68 |

Note: Parameters carrying (*) are the algorithm's default values and (**) indicates the results for optimized parameters where user-defined parameters dt stand for class threshold, GR for grid resolution, and ts for the time step of the CSF algorithm.

Similarly, Table 5 shows that the PMF default parameter values with w = 1.5 and Δh = 0.30, the evaluation metrics of T.I., T.II., T.E., and k, were 0.00, 44.87, 28.02, and 47.91, respectively. Compared with CSF, the PMF error budget was also found to be higher at default values. The optimized user-defined parameters with w = 2, and Δh = 0.10 obtained by sequential adjustment resulted in the T.I., T.II., TE., and k values of 12.01, 07.68, 09.62, and 84.37, respectively. Nevertheless, the extreme adjustment in default parameters, such as w = 2 and Δh = 0.05, resulted in T.I., T.II., T.E., and k values of 45.93, 1.80, 18.36, and 57.23, respectively. For CSF and PMF, the results are indicative of no improvement with an extreme adjustment approach (Tables 4 and 5).

**Table 5.** PMF ground classification error metrics and kappa coefficients for different parameter values.

| Site ID | Method | Parameters | | Results | | | |
|---------|--------|------|--------|-----------|-----------|-----------|-----------|
| | | w | Δh | T.I. (%) | T.II. (%) | T.E. (%) | k (%) |
| A | PMF | 1.5 * | 0.30 * | 0.00 | 44.87 | 28.07 | 47.91 |
| | | 1.3 | 0.30 | 0.00 | 45.84 | 28.67 | 46.93 |
| | | 1.3 | 0.20 | 0.00 | 36.11 | 22.18 | 56.45 |
| | | 1.3 | 0.10 | 3.32 | 13.33 | 09.59 | 80.36 |
| | | 2 | 0.10 | 12.01 | 7.68 ** | 9.62 ** | 84.37 ** |
| | | 2 | 0.05 | 45.93 | 01.80 | 18.36 | 57.23 |

Note: Parameters carrying (*) are the algorithm's default values and (**) indicates the results for optimized parameters, where w stands for window size and Δh for elevation differences threshold of minimum and maximum surfaces of the PMF algorithm.

Tables 4 and 5 indicate that the CSF and PMF require default parameter optimization through sequential adjustment of one or more default parameters to achieve the best ground classification results for UAS-LiDAR data. Furthermore, one parameter adjustment while keeping the remaining parameters constant is an effective approach as demonstrated in this study.

PointCNN first was trained and validated using training and validation datasets as presented in Section 2.3 with an overall validation accuracy of 90, as shown in Figure 8, which is in line with the kappa coefficient of 90.18 obtained for test Site A (Table 6). Consistent error metrics of T.I., T.II., and T.E. with values 05.30, 04.26, and 04.66 quantify the good performance of PointCNN over training Site A (Table 6). To improve the PointCNN performance over training Site A, different learning rates along with an adjustment in the total epochs were also investigated. Table 6 indicates that PointCNN performance by changing the learning rate along with the number of epochs did not improve the accuracy. Nevertheless, PointCNN was found to perform better than ground classification algorithms over test Site A; the transferability potential of ground classification algorithms and PointCNN is analyzed in Section 4.3.

**Table 6.** PointCNN ground classification error metrics and kappa coefficients.

| Site ID | Method | Parameters | | | | Results | | | |
|---------|--------|------------|--------|-----------|---------------|----------|-----------|----------|----------|
| | | Batch size | epochs | Iteration | Learning rate | T.I. (%) | T.II. (%) | T.E. (%) | k (%) |
| A | PointCNN | 4 | 100 | 18,300 | 0.003 | 5.30 ** | 4.26 ** | 4.66 ** | 90.18 ** |
| | | 4 | 150 | 36,600 | 0.004 | 09.68 | 07.42 | 08.63 | 89.20 |
| | | 4 | 200 | 27,450 | 0.002 | 12.71 | 10.02 | 10.20 | 67.77 |

Note: Values carrying (**) indicate the best performance of PointCNN.

### 4.3. Transferability Analysis

The transferability assessments of CSF and PMF are accomplished by processing the test sites (Figure 2) using the optimized user-defined parameters (see Tables 4 and 5). To the DL framework end, the test sites were processed using the best-trained PointCNN model (Section 3.2.1 and Table 6).

Figure 11 illustrates that among the ground classification algorithms, the CSF algorithm shows consistent kappa coefficients and lower error metrics of T.I., T.II., and T.E., compared with the PMF algorithm. T.I. error was found to be significantly lower than T.II. error for the CSF algorithm overall study sites. By contrast, the error metrics of PMF were found to be inconsistent and significantly higher than the CSF algorithm. Particularly, sites A, C, and D show significantly higher rates of T.I., T.II., and T.E. errors for the PMF algorithm compared with CSF and PointCNN (Table 7).

To understand the underlying effect of PMF lower performance, the characteristics of the acquired UAS-LiDAR data were further considered (Table 2). It was found that, in terms of UAS-LiDAR altitude (AGL), pulse reputation rate PRR (kHz), and corresponding point densities, the PMF performance was found to be sensitive toward the point densities of UAS-LiDAR data. Table 2 shows that the point densities of Site A, B, C, D, and E were 590.82 pts/m$^2$, 390.43 pts/m$^2$, 593.50 pts/m$^2$, 903.89 pts/m$^2$, and 484 pts/m$^2$ with the PMF corresponding kappa coefficients of 84.37, 93.05, 75.70, 68.27, and 89.72, respectively. The results indicate a general trend of lower kappa coefficients of PMF with increasing point densities (Tables 2 and 5). For example, for Site C with a point density of 593.89 pts/m$^2$, a kappa value of 75.70 was found. Similarly, for Site D with the highest point density of 903.89 pts/m$^2$, PMF yields a kappa value of 68.27, which indicates an all-time low kappa coefficient (Table 7). On the contrary, the CSF has shown consistent kappa coefficients at all point densities for training and test sites (Figure 11 and Table 7). The present study indicates that the CSF performance with UAS-LiDAR data was found to be better than the PMF algorithm (Figure 11 and Table 7).

Figure 11 displays that compared with the ground classification algorithm, PointCNN gives highly correlated kappa coefficients with the least error metrics of T.I., T.II., and T.E. for Sites A, B, D, and E compared with the PMF and CSF algorithms. However, for Site C, the CSF performance is slightly better with a kappa coefficient of 84.94 compared with the PointCNN kappa coefficient of 84. 89. Moreover, the PointCNN T.I., T.II., and T.E. errors were found to be higher for Site C and comparable with the CSF algorithm.

The overall performance indicates that PointCNN outperforms the traditional ground filtering algorithms, yet Site A and Site C require further explanation of higher error metrics and kappa coefficients for the PointCNN model.

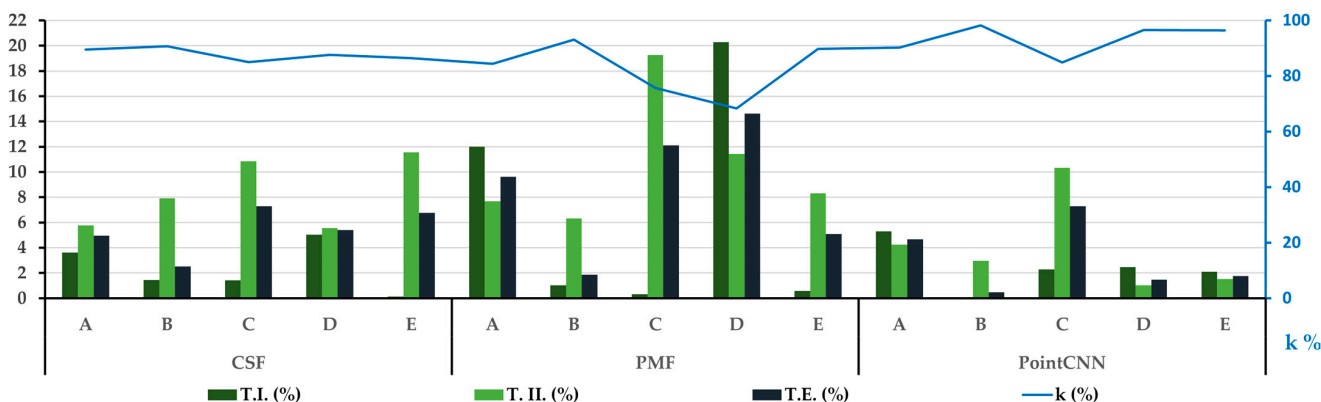

**Figure 11.** Comparative graph plot of error metrics of T.I., T.II., and T.E. with kappa coefficients (k) of CSF, PMF, and PointCNN for training Sites A, and test sites B, C, D, and E.

**Table 7.** Transferability evaluation of CSF, PMF, and PointCNN for test sites with error metrics and kappa coefficients.

| Method | Site ID | Results | | | |
|---|---|---|---|---|---|
| | | **T.I. (%)** | **T.II. (%)** | **T.E. (%)** | **k (%)** |
| CSF | B | 1.46 | 7.93 | 2.51 | 90.75 |
| | C | 01.42 | 10.84 | 07.29 | 84.94 ** |
| | D | 5.04 | 5.56 | 5.40 | 87.64 |
| | E | 0.14 | 11.57 | 6.77 | 86.38 |
| PMF | B | 1.02 | 6.31 | 1.88 | 93.05 |
| | C | 0.31 | 19.25 | 12.12 | 75.70 |
| | D | 20.28 | 11.44 | 14.63 | 68.27 |
| | E | 0.59 | 8.32 | 5.08 | 89.72 |
| PointCNN | B | 0.00 | 2.98 | 0.48 | 98.19 ** |
| | C | 2.30 | 10.34 | 7.30 | 84.89 |
| | D | 2.46 | 1.04 | 1.48 | 96.53 ** |
| | E | 2.11 | 1.52 | 1.77 | 96.36 ** |

Note: Values carrying (**) indicate the best performance of the following method.

## 5. Discussion

This section of the paper discusses the overall performance of ground classification algorithms and the PointCNN model to shed light on several important aspects of the used methods in comparison with the quality of the UAS-LiDAR data acquired under different crop environments (Figure 2).

Figure 11 and Table 7 illustrated that the overall PointCNN has been proven to outperform the traditional algorithms in the context of classification and transferability of ground points in five representative agricultural site areas. However, the overall quantitative analysis of ground classification algorithms and PointCNN using error metrics of T.I., T.II., T.E., and k were found to be significantly different for training Site A, as given in Tables 4–6, compared with test sites B, D, and E, given in Table 7.

We found that all methods are subject to the rationality of ground and non-ground features present in UAS-LiDAR data. For illustration, training Site A presents a more diverse crop environment with five different crops (see Section 2.1.1), and dry soybean presents a complex back-scattered UAS-LiDAR signal, as shown in Figure 12c. Past studies showed that low-cost UAS-LiDAR sensors suffer from point localization uncertainty over complex surfaces [5,29]. Therefore, complex back-scattered UAS-LiDAR signal is challenging over certain crop environments to differentiate ground from non-ground points (Figure 12c, brown-box). Nevertheless, the problem is associated with the sensor capabilities and high-end high-cost UAS-LiDAR sensors (e.g., http://www.riegl.com/products/

unmanned-scanning/riegl-vux-1uav (accessed on 4 January 2023)), therefore offering a better solution [5]. The crops where point clouds present reasonable patterns, i.e., spatially local correlation [39], were classified correctly (Figures 10 and 12d,e). Because of the dry soybean in Site A (Figure 12c), the overall accuracy of CSF, PMF, and PointCNN is lowered compared with test sites B, C, D, and E (see Figure 11 and Table 7).

Ground classification algorithms are based on simulating the overall surface from point patches (e.g., CSF) or interpolating the minimum and maximum or mean values from point patches (e.g., PMF, MCC) with several default parameters [18,49]. The complex point cloud patches of dry soybean (Figure 12a–c) have proven to be challenging for the ground classification algorithms as data captured by ZENMUSE L1 (Tables 6 and 7) are of low quality due to point localization problems [5,28]. The 10% validation loss with overall 90% validation accuracy of PointCNN (Figure 8) is caused by the dry soybean of Site A (Figure 12). Compared with PMF, the CSF provides more control over the simulated surface by adjusting default parameters such as GR, RI, dt, and the total number of iterations, therefore resulting in better accuracy given the fact that it is a simulation-based method more advanced compared with PMF, which is an old-fashioned interpolation based method [35].

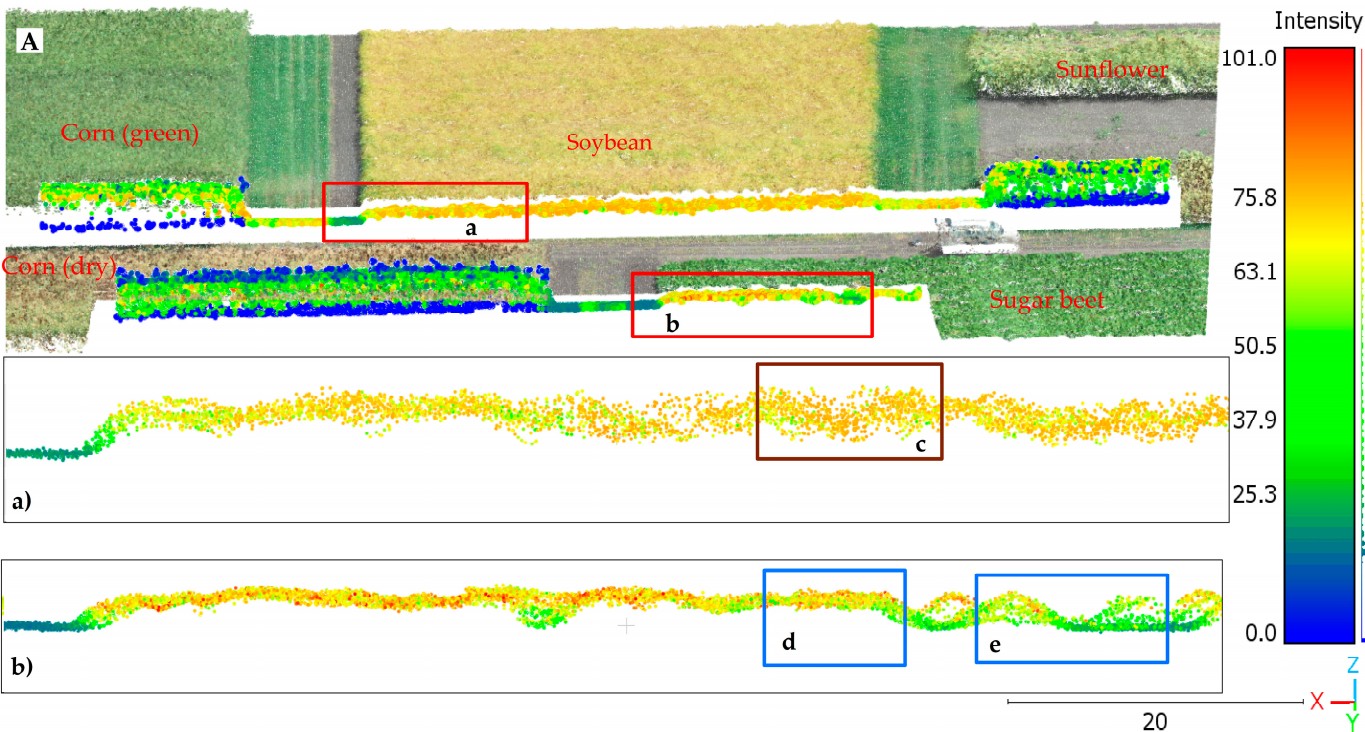

**Figure 12.** (**A**) Colored DJI L1 point clouds (Site A) with 3D transects of intensity information: (**a**) Transect representative of soybean, and (**b**) sugar beet. (**c**) Complex backscattered UAS-LiDAR point clouds of dry soybean, (**d**) represent sugar beet with only non-ground returns from the top of the canopy, and (**e**) sugar beet with canopy and ground returns. Three-dimensional transects (**a**–**e**) are of the same extent (horizontal) for visualization purposes with no vertical scale.

The transferability assessment over test sites yields lower error metrics with a consistent kappa coefficient for PointCNN compared with traditional ground classification algorithms. On the contrary, Site C is the only test site where the PointCNN performance is slightly lower than the CSF, as depicted in Figure 11 and Table 7. The PointCNN's relatively lower performance compared with CSF is further studied for Site C.

PointCNN treats low-lying flat crops as the ground point compared with the CSF algorithm (Figure 13b–c). The CSF with a cloth rigidness value of three puts more restrictive conditions on the simulated surface, resulting in better segregation of low-lying vegetation from the ground (Figures 5 and 13c) [35]. As a result, a significant portion of Site C, which is

enclosed with a yellow box in Figure 13A, is of a predominantly unique crop environment, where CSF performs slightly better than PointCNN. The crop environment enclosed in the yellow box (Figure 13A) has a greater resemblance with the flat surface; therefore, PointCNN treats those points as ground points.

Figures 13 and 14, as exemplary test site cases, illustrate that the remaining test sites (B and E) were also comprised of growing crops with a higher degree of chlorophyll content compared with training site A (Figure 12). The LiDAR backscatter signal is stronger for test sites given the fact that healthy vegetation tends to reflect more in the infrared portion of the electromagnetic spectrum because LiDAR sensors generally operate in infrared regions [62]. In addition, UAS-LiDAR has a much smaller footprint than ALS systems; therefore, almost the entire LiDAR backscatter returned either from the top of the crop (e.g., leaves) or the ground; therefore, ground and non-ground points were captured with better separation over test sites [5]. On the contrary, dry crops with low chlorophyll content particularly low-lying crops, e.g., dry soybean in Site A (Figure 12a), established a complex backscattered signal, making it challenging for ground classification algorithms and PointCNN to segregate ground from non-ground points at certain positions (Figure 12c).

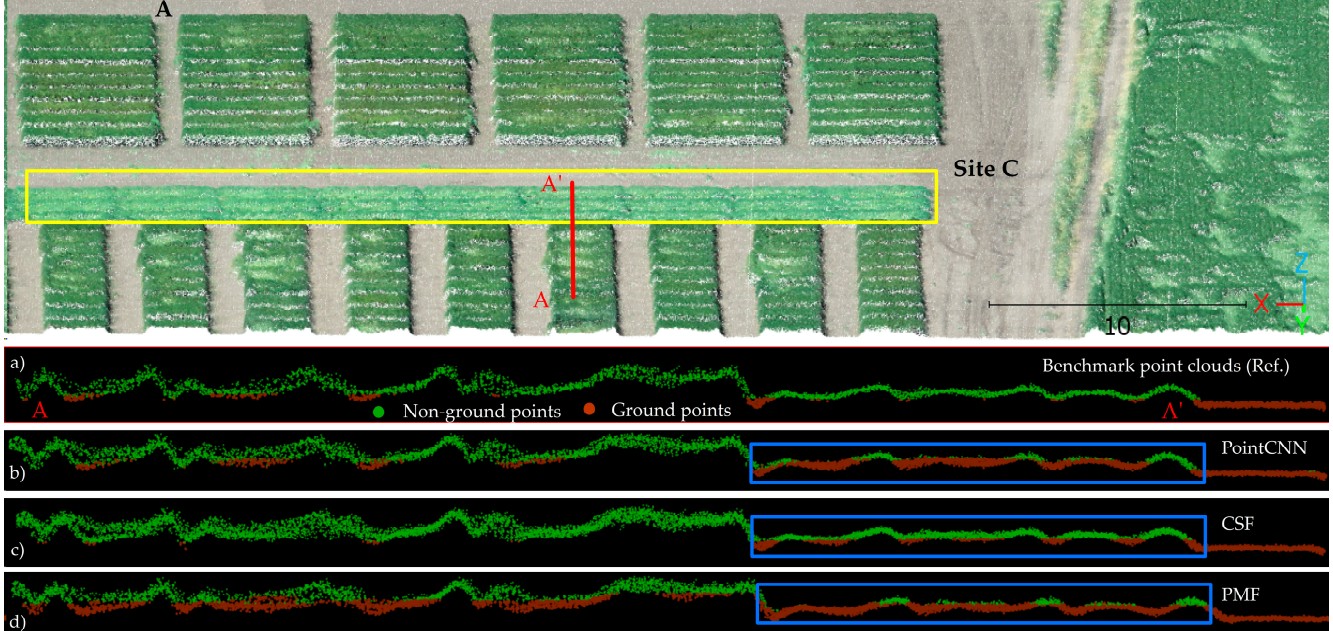

**Figure 13.** (**A**) Colored DJI L1 point clouds (Site C), 3D transects (AA′) of ground and non-ground points: (**a**) Transect representative of pulse-crop with benchmark point clouds. Ground and non-ground points are filtered by (**b**) PointCNN, (**c**) CSF, and (**d**) PMF. AA′ transects (**a**–**d**) are of the same extent (horizontal) for visualization purposes with no vertical scale.

One of the objectives of the present investigation is to assess the PointCNN and ground classification algorithm's potential to classify ground points from ditches of an undulating terrain of Site D (Section 2.1.4). Figure 14 shows the ground points classification results of PointCNN, CSF, and PMF as an example case. PointCNN and CSF do not take ground points into account along the slope surfaces of the ditch (Figure 14b–c). The PMF overall performance for Site D was found to be poor (Figure 11, Table 7). On the contrary, PMF ground classification along the slope surface is better compared with CSF and PointCNN (Figure 14d).

The PMF algorithm filters the ground points based on the minimum elevation differences between a minimum surface and maximum surface generated from point clouds, therefore undulating topography does not affect its performance (Section 3.1.2) [20]. However, CSF is based on a simulated cloth with rigidness values of three putting more tension

on the simulated surface; therefore, ground points from the bottom of the ditch are only extracted as simulated cloth over inverted point clouds does not take the slope factor into account (Section 3.1.1) [51]. Regarding the CSF, the ground points classification accuracy can be improved by enabling the slope default parameter available in the CSF algorithm [35]. Regarding the DL, PointCNN was trained with point clouds representative of the flat terrain (Figure 2a); therefore, it performs poorly along the ditches (Figure 14b) located in agricultural fields of test Site D. Nevertheless, PointCNN performance over undulating terrain is a subject of further investigations, which is beyond the scope of the present study. In the context of ground points classification at early growth stages of Site E, the qualitative analysis in Section 4.1 and quantitative analysis in Section 4.2, showed that PointCNN is robust in classifying ground points from crops at early growth stages along with grass and weeds, as already demonstrated in Figure 11 and Table 7.

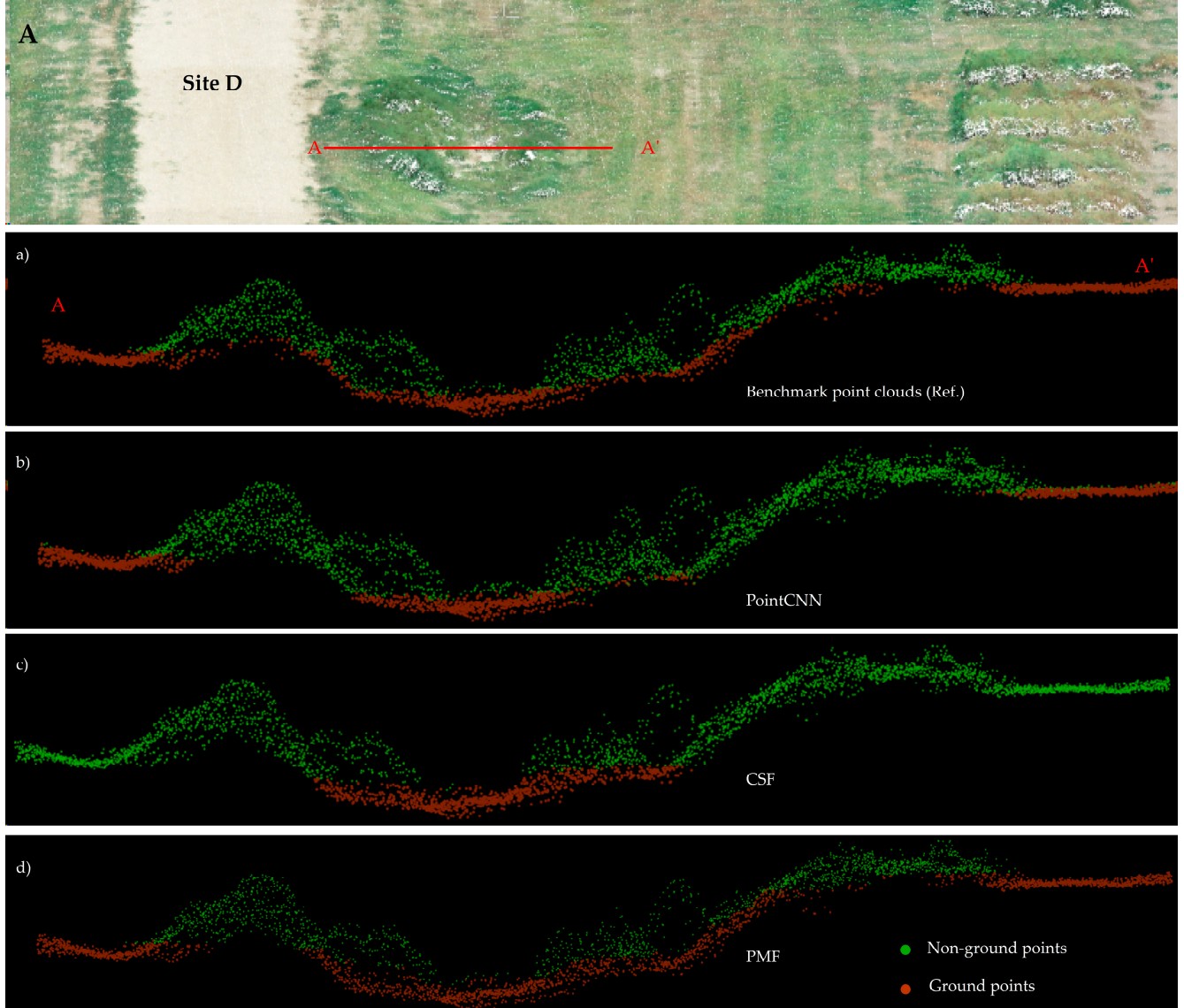

**Figure 14.** (**A**) Colored DJI L1 point clouds (Site D) with 3D transects (AA′) of ground and non-ground points: (**a**) 3D transects representative of a ditch with benchmark point clouds. (**b**) ground and non-ground points filtered by (**b**) PointCNN, (**c**) CSF, and (**d**) PMF. AA′ transects (**a**–**d**) are of the same extent (horizontal) for visualization purposes with no vertical scale.

The present study, in the context of classification and transferability using error metrics of T.I., T.II., T.E., and k coefficients, reveals that PointCNN outperforms CSF, PMF, MCC, and ArcGIS ground classification algorithms. Fundamentally, the ground classification algorithm's performance is subject to user-defined parameters that were proven to be sensitive toward UAS-LiDAR data or terrain characteristics, in particular, the low-lying vegetation [63]. On the contrary, the DL framework of PointCNN is a data-driven approach sensitive towards training data quality and overall feature representation in UAS-LiDAR point clouds, therefore, little affected by the different and diverse crop environments [64].

## 6. Conclusions

Automated bare earth (ground) point classification is an important and most likely step in LiDAR point cloud processing. In the past two decades, substantial progress has been made and many ground classification algorithms have been developed and investigated, generally focusing on forest and topographic research using ALS point clouds. In comparison with matured scientific research in forests and topographic domains, the UAS-LiDAR sensor that has recently emerged has not yet been investigated, particularly in the precision agriculture domain. In this paper, we presented the first evaluation of the frequently used ground classification algorithms (CSF, PMF, and MCC) and PointCNN using UAS-LiDAR data in an agricultural landscape.

The present study aimed at two aspects of ground classification algorithms. First, the authors provided default user parameters of ground filter algorithms, which were examined, and higher omission and commission errors were found for time-series UAS-LiDAR data. Furthermore, our investigation revealed that sequential adjustment of the algorithm's default parameters showed overall better results for CSF and PMF algorithms; therefore, default parameter optimization is important for new sensors and data. Second, we assessed the transferability potential of the ground classification algorithm with optimized parameters compared with the Deep Learning framework of PointCNN over four test sites each of unique crop environments in North Dakota, USA. Our investigation showed that the ground classification algorithms inherit the transferability potential with the optimization of default parameters, as tested in this study, with overall reasonable error metrics and kappa coefficients over four representative agricultural plots. Nevertheless, PointCNN was found to be more robust in both contexts, i.e., overall accuracy and transferability, as it showed the least error metrics and consistent kappa coefficients compared with traditional ground classification algorithms.

Considering experimental results and discussion regarding the UAS-LiDAR ground points classification, several key findings can be concluded. First, to use the ground classification algorithms, parameter optimization is required through sequential adjustment in parameter default values. Algorithm parameters presented in this study can be used to derive better ground classification results using UAS-LiDAR data. Second, compared with traditional ground classification algorithms, deep learning methods are robust with proven accuracy and efficiency provided that high-quality accurate label data are available. In conjunction with PointCNN supervised classification, the unsupervised ground classification algorithms can be used to process the raw point clouds as a first step toward producing quality labeled data through some manual interventions.

**Author Contributions:** Conceptualization, N.F. and J.P.F.; methodology, N.F.; software, N.F., J.P.F. and A.K.D.; validation, N.F. and J.P.F.; formal analysis, N.F.; investigation, N.F.; resources, J.P.F.; data curation, N.F. and A.K.D.; writing—original draft preparation, N.F.; writing—review and editing, J.P.F. and N.F.; visualization, N.F.; supervision, J.P.F.; project administration, J.P.F.; funding acquisition, J.P.F.; All authors have read and agreed to the published version of the manuscript.

**Funding:** This research was funded by the United States Department of Agriculture (USDA), Agriculture Research Service (ARS), under agreement No. 58-6064-8-023 and The APC was funded by USDA-ARS agency names at https://www.ars.usda.gov/ (accessed on 12 January 2023).

**Data Availability Statement:** The data used in this research are available upon request to the corresponding author. The trained PointCNN model is available at https://sites.google.com/ndsu.edu/floreslab/tools?pli=1 (accessed on 12 January 2023).

**Acknowledgments:** Authors thank the anonymous reviewers for their careful reading of the article and for providing valued suggestions, detailed insight, and recommendations. The insight provided by reviewers brought substantial improvement in the article, which otherwise would not have been possible. We also acknowledge and thank USDA-ARS for the financial support for this research.

**Conflicts of Interest:** The authors declare no conflict of interest.

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
