# Peer review of "Analysis of UAS-LiDAR Ground Points Classification in Agricultural Fields Using Traditional Algorithms and PointCNN"

_remotesensing, doi:10.3390/rs15020483_

Round 1

Reviewer 1 Report

1.      The advantages and disadvantages of UAS-LiDAR and UAS-red-green-blue (RGB) sensors should be discussed in Line 39.

2.      The introduction is not detailed enough. At present, UAS-RGB has been widely used in the study of crop height estimation and ground feature classification. Compared with UAS-RGB, UAS-LiDAR could minimize the interference of light intensity, instead of Increasing point cloud density.

3.      Line 93-95 point that scientific investigations on ground point classification using geometrically complex UAS-LiDAR point clouds have not yet been conducted. As far as I know, there have been many research on ground point classification using geometrically complex UAS-LiDAR point clouds.

4.      The types of the training set and the test set in Table 3 are different. Please check.

5.      50% forward and lateral overlap in line 173 is too low. high density point cloud data cannot be obtained.

6.      ArcGIS is a software that is not suitable as a title in line 289.

7.      Authors analyzes several common point cloud classification algorithms and optimizes some parameters. However, the algorithm has not been improved, and the innovation points are insufficient.

8.      The key point of using point cloud to classify ground objects is to achieve higher classification accuracy with less point cloud data. Different line overlap and UAV flight height have great influence on the precision of point cloud density extraction. It is suggested that the authors add experiments and improve the algorithm to achieve better results.

Author Response

Authors thanks the reviewers for their valuable time and suggestions. We tried our best to understand the reviewers’ comments and address them to per best of our knowledge. However, if there is any confusion while addressing the comments and suggestions, we duly mentioned them in the response.  Please read the point-by-point response as in the attached PDF.

Reviewer 2 Report

Automatic ground classification is an integral part of LiDAR point cloud processing. The work was based on enough review of the relevant literature on the subject. The work is an extensive study of cases using methods or algorithms of classification of ground points such as CSF, PMF, MCC and PointCNN from UAS-LiDAR data.

The authors used measured data from several test fields and conducted detailed empirical performance analyses of individual algorithms - optimizing their parameters. The number of these calculations is impressive.

I consider using only ready-made tools (implementation of application algorithms) as a disadvantage of this article. This approach only guarantees partial knowledge of the specificity of the algorithm and the definition of their bottlenecks, which in the article are assessed, for example, by indicating a long time of calculations. This time does not have to result from the disability of the algorithm but, for example, from its faulty implementation. The use of ready implementation also does not allow their innovative optimization beyond interference at input and output points.

The Authors used the algorithms' default values and tried to find their optimised values in the iterative process, thus giving all to better values. These values can be suitable for specific test fields. The presented conclusions do not convince the possibility of using the presented methodology as a universal parameterisation of algorithms.

It should be pointed out that some of the algorithms used are a kind of “black box” (e.g. from ARC GIS) and it is difficult to evaluate them all the more after the effects.

75% of content of the article is a short description of existing algorithms. This description is not enough to understand their functionality – some one who doesn’t know them must reach for source materials. Hence, it is doubtful to place their description in this form.

The novelty that you can find is introducing a new method to automate point-wise omission and commission errors using algorithm-based point clouds compared with benchmark point clouds. This is not enough to consider such an extensive article as innovative.

Author Response

Authors thanks the reviewers for their valuable time and suggestions. We tried our best to understand the reviewers’ comments and address them to per best of our knowledge. However, if there is any confusion while addressing the comments and suggestions, we duly mentioned them in the response.  Please read the point-by-point response in attached PDF.

Reviewer 3 Report

The paper presents in a very accurate way the use of the DJIL1-LiDAR in the determination of terrain in agricultural land, which not only helps to determine the terrain surfaces, but will also improve crop classification techniques. This type of study, as the authors rightly say, was pending a rigorous evaluation, which the authors do meritoriously in this paper.

 The Research is very well formulated, with an initial part of the study of very extensive calculation methodologies and a rich number of case studies. This reviewer values the article very positively and would like to make some comments to assess its authors:

- In the work, the authors use a first Site A as a training base that they later take to the rest of the zones. The authors themselves analyze whether this greater variability in Site A will not be a matter of controversy when extrapolating the training results to the other Sites. It would not have been better to establish two subtypologies for training in Site A. Was there any analysis in that direction throughout the work? Do you think it's worth including?

- The authors raise in line 601 and perhaps other places in the article, the possibility that a lower precision in the determination of the surfaces has led to worse yields in the results. It is not clear to the reviewer if this discussion or conclusion is due to lack of precision in the LiDAR equipment used or low precision obtained in determining the study surfaces. In fact, the reviewer states:

o Could it be that a previous precision study of the point clouds will help to improve the typification of the surfaces and that introducing these parameters as part of the study algorithms will improve the results?

o It may be that in low crop heights, the possible noise in the determination of the point cloud is part of some of the worst results (compared to others) obtained in low crops as indicated at some point in the article (line 720) and could this influence the differentiation in the comparative results between the proposed algorithms?

The reviewer raises with these comments to know if it is possible to improve the text by reviewing their comments.

Apart from these comments, the reviewer proposes the revision of two points of the article:

- In line 172, it is indicated that the Drone used was the Matrice 200 RTK and perhaps it was intended to indicate that it was the Matrice 300RTK

- They could indicate that although there is no vertical scale in the figures with crop profiles, they do maintain all profiles the same scale so that comparisons in the representation of point clouds or determined surfaces are possible.

Author Response

Authors thanks the reviewers for their valuable time and suggestions. We tried our best to understand the reviewers’ comments and address them to per best of our knowledge. However, if there is any confusion while addressing the comments and suggestions, we duly mentioned them in the response.  Please read the point-by-point response in the attached PDF.

Round 2

Reviewer 1 Report

The sensor is only a means to obtain data, and the algorithm structure should be improved rather than simply optimizing parameters

Author Response

Dear reviewers,

Thank you very much for your constructive feedback and evaluation of our article. Please check our response to your feedback and comments.

"The sensor is only a means to obtain data, and the algorithm structure should be improved rather than simply optimizing parameters"

The title of our article is "Analysis of UAS-LiDAR Ground Points Classification in Agricultural Fields using Traditional Algorithms and PointCNN" 

Therefore, development in ground classification algorithms is not the objective of our research. In the context of modifying an existing algorithm a sole publication is required to address this topic with extensive research and development, again which is not the objective of the present research. Let us summarize our research objectives again.

(a) analysis of ground classification algorithms in two contexts

(i) using UAS-LiDAR high-density point clouds

(ii) focusing on an agricultural context, which is missing from existing research

We assessed the performance of the most widely used ground classification algorithms and the only thing that can be improved at user-ends is to optimize the default parameters for new sensor data and a new environment i.e., an agricultural field.

(b), transferability assessment of ground classification algorithms compared with deep learning PointCNN model.

First, the research questions being asked in the research objective have been thoroughly addressed in the article. Therefore, we value your suggestions and recommendation but improving the functionality and structure of an existing ground classification algorithm is beyond the scope of the present study. Secondly, this article categorically says "deep learning is robust" compared with traditional ground classifications algorithms, therefore, the authors' future research focuses on deep learning rather than improving existing ground classification algorithms.

Thank you so much for your valuable suggestions and comments that helped us to improve the present article.

Reviewer 2 Report

Explanations presented by the author's convince me. Finally, I assess the article as a valuable for publication.

Author Response

Dear reviewer,

Thank you so much for providing us with valuable suggestions and recommendations.

Reviewer 3 Report

Authors have made all corrections proposed by this reviewer.

I would like to thanks for their job and congratulations for the paper.

Author Response

(The authors gave the same response as above.)
